# Opportunities for improved surveillance and control of dengue from age-specific case data

Isabel Rodriguez-Barraquer[1]*, Henrik Salje[2,3,4,5,6], Derek A Cummings[6,7]

[1]Department of Medicine, University of California, San Francisco, San Francisco, United States; [2]Mathematical Modelling of Infectious Diseases Unit, Institut Pasteur, Paris, France; [3]CNRS, URA3012, Paris, France; [4]Center of Bioinformatics, Biostatistics and Integrative Biology, Institut Pasteur, Paris, France; [5]Department of Epidemiology, Johns Hopkins Bloomberg School of Public Health, Johns Hopkins University, Baltimore, United States; [6]Department of Biology, University of Florida, Gainesville, United States; [7]Emerging Pathogens Institute, University of Florida, Gainesville, United States

**Abstract** One of the challenges faced by global disease surveillance efforts is the lack of comparability across systems. Reporting commonly focuses on overall incidence, despite differences in surveillance quality between and within countries. For most immunizing infections, the age distribution of incident cases provides a more robust picture of trends in transmission. We present a framework to estimate transmission intensity for dengue virus from age-specific incidence data, and apply it to 359 administrative units in Thailand, Colombia, Brazil and Mexico. Our estimates correlate well with those derived from seroprevalence data (the gold standard), capture the expected spatial heterogeneity in risk, and correlate with known environmental drivers of transmission. We show how this approach could be used to guide the implementation of control strategies such as vaccination. Since age-specific counts are routinely collected by masany surveillance systems, they represent a unique opportunity to further our understanding of disease burden and risk for many diseases.
DOI: https://doi.org/10.7554/eLife.45474.001

*For correspondence:
lsabel.Rodriguez@ucsf.edu

**Competing interests:** The authors declare that no competing interests exist.

## Introduction

A fundamental challenge of disease surveillance systems is how to transform data that is routinely collected into useful, actionable evidence that can inform control interventions. Disease surveillance systems typically focus on analyzing aggregate counts of cases over defined time periods to stratify the risk of populations (*WHO, 2012*). However, the use of raw case counts, or incidences, as a measure of disease risk can frequently be misleading because the quality of surveillance often differs significantly both across countries and within regions of a country making comparisons inappropriate. Areas with more complete reporting of disease may inaccurately appear to have more disease simply because reported cases scale linearly with completeness of reporting. Surveillance systems may change over time (e.g. improving in completeness over time) making comparisons difficult or even impossible. These problems are particularly troubling when examining disease trends or ranking regions according to their disease risk. Recent efforts have tried to improve quantification of disease burden by pooling numerous sources of data. For example, disease mapping methods that combine disease presence/absence data, environmental covariates and available incidence data (from cohort or cross-sectional studies) have been used to predict spatial limits and global case counts for diseases including malaria, dengue and several neglected tropical diseases. (*Hay et al., 2009*;

*Bhatt et al., 2013*; *Pigott et al., 2014*; *Hay et al., 2013*) However, while these methods are successful in identifying boundaries of endemic areas, the robustness of these approaches to quantify transmission within endemic areas has not been validated.

For most immunizing infections (infections that generate long-lasting sterilizing immunity), serological surveys are regarded as the gold standard to measure the susceptible fraction and infer the extent of transmission, as they provide a direct measure of the proportion of the population that has been infected. This is particularly useful for diseases such as dengue, influenza or Zika, where the asymptomatic to symptomatic ratios are large or unknown. Methods to estimate transmission parameters from age-stratified serological data have been available for many years(*Muench, 1959*) and have been used to analyze trends in transmission for multiple diseases including measles (*Grenfell and Anderson, 1985*), hepatitis A(*Schenzle et al., 1979*), dengue (*Ferguson et al., 1999*; *Imai et al., 2015*; *Rodriguez-Barraquer et al., 2011*; *Rodríguez-Barraquer et al., 2014*), pertussis (*Kretzschmar et al., 2010*), influenza (*Miller et al., 2010*), malaria (*Bretscher et al., 2013*) and chikungunya (*Salje et al., 2016*). While well conducted age-stratified serological surveys can provide an unbiased measure of the susceptibility profile of a population and transmission parameters, they are generally not part of routine surveillance activities and are therefore only available for a limited number of locations and time points. For example, an extensive review of the dengue literature published recently found only three population-based serosurveys conducted in Brazil over the past 10 years, despite being the country that currently reports the largest number of cases worldwide (*Fritzell et al., 2018*). Similarly, only one and three studies were identified for the Philippines and Thailand, respectively, even though the burden of dengue is very high (*Fritzell et al., 2018*). Thus, the picture provided by serology is very incomplete and limited when trying to characterize transmission across time and space both within and between countries.

While aggregated case counts can be misleading when quantifying disease risk, the age distribution of incident cases contains a lot of information on the age-specific susceptibility of the population. Importantly, the age distributionof cases is also largely robust to under-reporting, facilitating the comparison between locations or over time. By combining age-specific incidence data and mechanistic models of how population immunity is acquired over time, it is possible to estimate key transmission parameters and obtain a much more accurate picture of the local and global burdens of disease. Since age-specific counts are routinely collected by surveillance systems as part of standard practice, they represent a missed opportunity to further our understanding of epidemic patterns for many diseases.

Here, we use dengue virus as an example to illustrate how age-specific incidence data can be used to quantify disease transmission and inform control interventions. Dengue is a relevant example because, despite being the most widely spread mosquito-transmitted virus, large gaps remain in our understanding of its global and local epidemiology (*Fritzell et al., 2018*). We present a model to estimate the transmission intensity of dengue from age-specific incidence data, and apply it to surveillance data from administrative units in four countries that suffer from endemic dengue transmission (Thailand, Colombia, Brazil and Mexico). We validate our estimates using serological data and show that they correlate well with known environmental drivers of dengue transmission at subnational level. Finally, we show how this approach could be used to guide the implementation of dengue control strategies such as vaccination.

## Results

We estimated the average forces of infection (FOI) over the last 20 years for 148 administrative level one units where age specific case data was available (*Table 1*, *Figure 1B*): 72/76 provinces of Thailand, 28/32 departments of Colombia, 21/27 states of Brazil, and 27/31 states of Mexico. These administrative units comprise 90%, 99%, 90% and 91% of the population at risk of these countries respectively. We also estimated forces of infection for 211 municipalities (administrative level two units) of Colombia where at least 200 cases were reported over the period covered by the data.

The average FOI (the rate at which susceptible individuals that will become infected in a year by any dengue serotype) was 0.096 in Thailand (95%CI 0.092-0.100), 0.132 (95%CI 0.128-0.136) in Colombia, 0.124 (95%CI 0.116-0.128) in Brazil, and 0.052 (95%CI 0.048-0.056) in Mexico. This implies that on average 9.6% of the susceptible population in Thailand gets infected every year by any of

**Table 1.** Data sources used

| Country | Years of data available | Type of report available | Type of report used | No. admin units analyzed* | Source of dengue data | Source of census data |
|---|---|---|---|---|---|---|
| Thailand | 1985–2010 | DHF | DHF | 72 | (*Bureau of Epidemiology, Department of Disease Control, 2019*) | (*National Statistical Office Thailand, 2017*) |
| Colombia | 2007–2012 | DHF/DF | All cases | 28/211 | (*Instituto Nacional de Salud Colombia, 2016*) | (*DANE, 2019*) |
| Brazil | 1998–2015 | DHF/DF† | DHF | 21 | (*Ministério da Saude, 2017*) | (*Instituto Brasileiro de Geografia e Estadistica, 2017*) |
| Mexico | 2000–2009 | DHF/DF | All cases | 27 | (*Direccion General de Epidemiología, 2017*) | (*Instituto Nacional de Estadistica y Geografia, 2016*) |

*Admin level one/Admin level two units with at least 150 cases reported

†Data from Brazil is on hospitalized dengue fever and dengue hemorrhagic fever cases.

DOI: https://doi.org/10.7554/eLife.45474.011

the circulating serotypes ($1 - e^{-FOI}$). Similarly, 12% of the susceptible population of Colombia and Brazil, and 5% of the susceptible population of Mexico get infected yearly.

However, as expected, transmission intensity varied greatly within countries, ranging between 0.04 and 0.15 (coefficient of variation (CV) = 0.27) across provinces of Thailand; between 0.02 and 0.20 across departments of Colombia (CV = 0.37), between 0.02 and 0.24 across states of Brazil (CV = 0.56); and between 0.01 and 0.092 across states of Mexico (CV = 0.45). Transmission was highest in the North East of Brazil (average FOI of 0.152) and in the Caribbean region of Colombia (average FOI of 0.156). There was also substantial heterogeneity within departments of Colombia. The mean CV for 15 departments where we had estimates for more than 5 municipalities was 0.4.

For the 16 locations where we had access to both age-stratified serological data (the gold standard) and case data, we found good correlation between the estimates of the FOI derived from both data sources (R2 = 0.73, 95% CI 0.51–0.87, *Figure 1C*). In contrast, we found no correlation between recent incidence of dengue in these locations (the average yearly incidence over the most recent 5 years of data) and the estimates of FOI derived from serological data (R2 = 0.002, *Figure 1—figure supplement 2*).

Since estimates of transmission intensity derived from seroprevalence data are only available for a small number of locations, to further validate our method we also explored the association between our estimates of the FOI for 211 Colombian municipalities (administrative level 2) with known environmental drivers of dengue transmission including temperature, elevation, precipitation a published metric of *Aedes aegypti* abundance and population density (*Figure 2*, *Figure 2—figure supplement 1*). Models were weighted by the number of cases used to estimate the FOI. On average, the FOI increased by 0.006 (95% CI 0.004–0.007, $R^2$ = 0.19) for each additional °C in temperature and, similarly, it decreased by 0.005 (95% CI 0.003, 0.006, $R^2$ = 0.21) for each 100 m increase in elevation (*Table 2*). While population density was not associated with FOI estimates in unadjusted analyses, a twofold increase in density was associated with a 0.007 (95% 0.004–0.009) increase in FOI in the best fitting adjusted model. This model included elevation, population density and precipitation, and explained 35% (95%CI 23–50%) of the variance.

In contrast, the recent incidence of dengue in these municipalities was not correlated with temperature, elevation or *Aedes aegypti* abundance (R2 0.01, 0.01, and 0.00 respectively, *Table 3*). We did find a negative association between population density and incidence, indicating a 6% (95%CI 2–10%) decrease in log incidence for each 2-fold increase in population density.

## Application: Guiding dengue vaccination policy

The only available dengue vaccine (Dengvaxia) has been licensed for use in children 9 years or older in 20 countries including Thailand, Brazil and Mexico. The WHO currently recommends confirmation (by virology or serology) of prior dengue infection at the individual level before vaccinating individuals, and therefore there is interest in identifying populations with high seropositivity to target pre-vaccination screening (*WHO, 2018*). In the absence of appropriate rapid serological assays that would allow implementing this individual screening strategy, an alternative that has been discussed is rolling out the vaccine in settings with 80% or greater seropositivity among 9 year olds. Using our

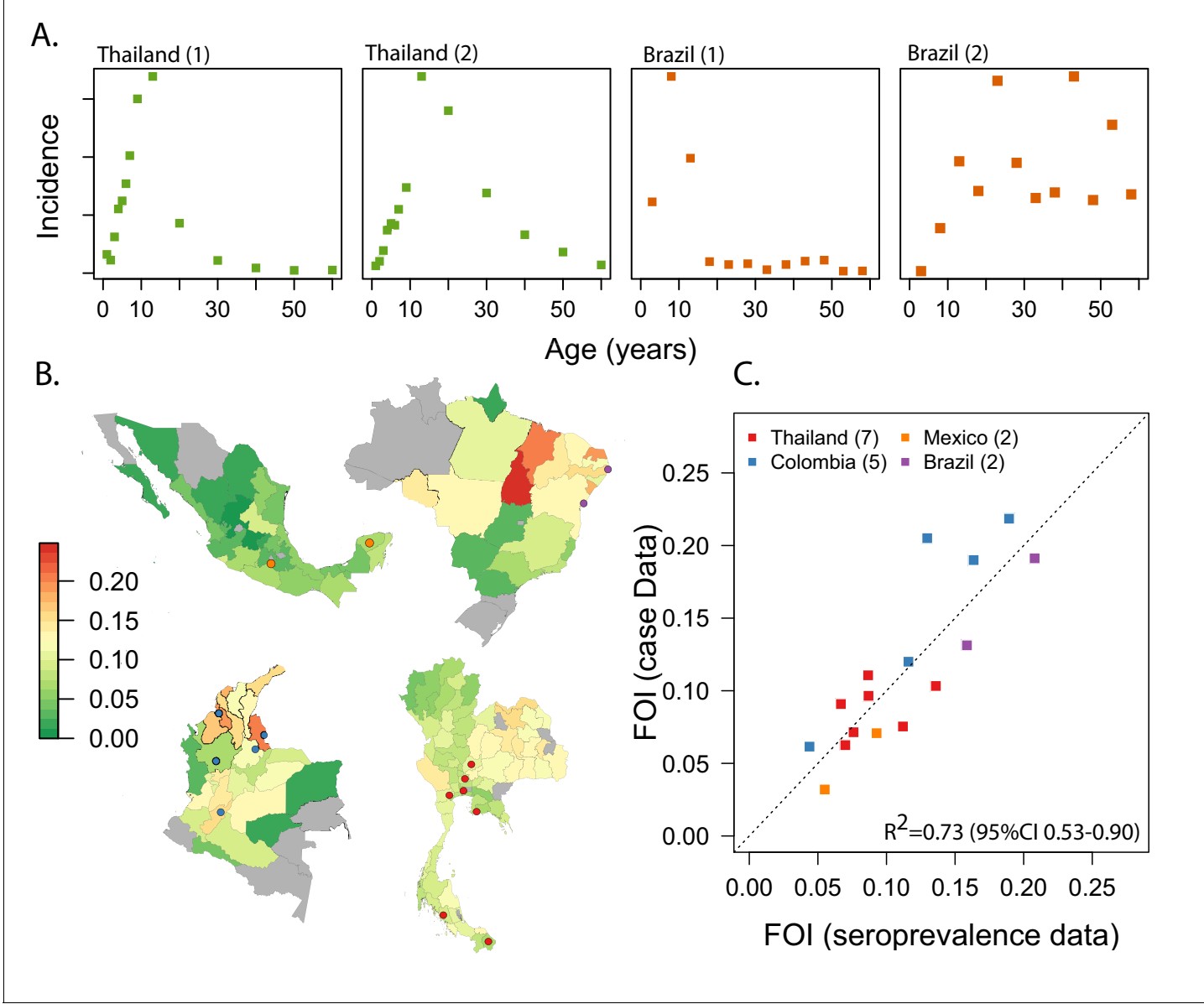

**Figure 1.** Estimating FOI from age-specific incidence data in Thailand, Colombia, Brazil and Mexico. (A) Examples of the age-specific incidence of dengue observed in two settings with very high endemic transmission (Thailand 1- Udon Thani, Thailand; Brazil 1 Pernambuco, Brazil) and two settings with lower and very low transmission (Thailand 2 = Chiang Mai, Thailand; Brazil 2 = Parana, Brazil). (B) Maps of our estimates of the FOI for the four countries. (C) Correlation between our estimates of the force of infection, with estimates derived from age-stratified serological data (gold standard) for 16 settings where we had both types of data (Thailand: Rayong (*Rodríguez-Barraquer et al., 2014*); Bangkok (*Imai et al., 2015*); Ratchaburi (*Imai et al., 2015*); Lop Buri, Narathiwat, Trang, Ayuttayah (*Vongpunsawad et al., 2017*). Brazil: Pernambuco (*Rodriguez-Barraquer et al., 2011*); Salvador (*Wilson et al., 2007*). Colombia (unpublished). Mexico: Morelos (*Amaya-Larios et al., 2014*), Yucatan (*Hladish et al., 2016*). The locations of the specific cities are shown in the maps in panel B.

DOI: https://doi.org/10.7554/eLife.45474.002

The following figure supplements are available for figure 1:

**Figure supplement 1.** Correlation between estimates of the FOI (derived from age-specific case data) and alternative metrics calculated directly from age-specific incidence data.

DOI: https://doi.org/10.7554/eLife.45474.003

**Figure supplement 2.** Correlation between estimates of the FOI derived from seroprevalence data (the gold standard) and several metrics derived from age-specific case data, for 16 spatial units where we had the two sources of data.

DOI: https://doi.org/10.7554/eLife.45474.004

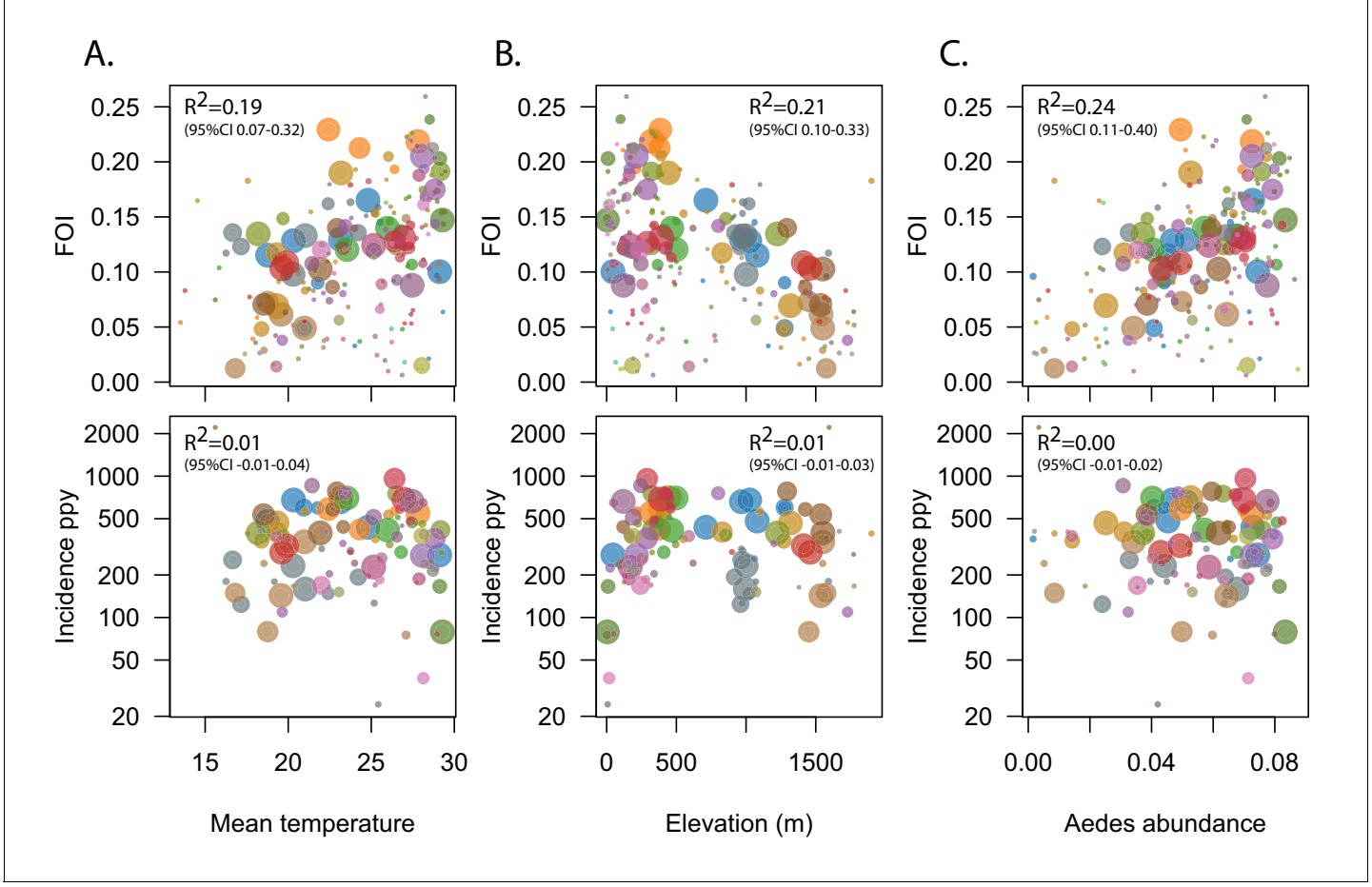

**Figure 2.** Correlation between estimates of the FOI and known environmental drivers of dengue transmission. Top panels show correlation between estimates of the FOI for 211 municipalities (administrative level 2) of Colombia and mean temperature (**A**), elevation (**B**) and Aedes abundance (**C**). Size of symbols is proportional to the number of cases available to estimate the FOI. Bottom panels shows lack of correlation between environmental drivers and recent incidence, the most commonly used metric of transmission intensity. $R^2$ values reported were obtained by fitting weighted linear regression models, with weights proportional to the number of cases used to derive the FOI estimate.

DOI: https://doi.org/10.7554/eLife.45474.005

The following figure supplements are available for figure 2:

**Figure supplement 1.** Correlation between estimates of the FOI and known environmental drivers of dengue transmission.

DOI: https://doi.org/10.7554/eLife.45474.006

**Figure supplement 2.** Heterogeneity in FOI within administrative level 1 units of Colombia.

DOI: https://doi.org/10.7554/eLife.45474.007

estimates of the FOI, we calculated the proportion of the population expected to be seropositive at

**Table 2.** Association between environmental variables and dengue force of infection for 211 municipalities in Colombia

| Variable | Unadjusted | | Adjusted | |
|---|---|---|---|---|
| | Estimate | 95% CI | Estimate | 95% CI |
| Elevation (per 100 m) | −0.005 | −0.003,−0.006 | −0.007 | −0.008,−0.006 |
| Mean temperature (per ˚C) | 0.006 | 0.004, 0.007 | | |
| Mean weekly precipitation | 0.0001 | −0.000, 0.001 | −0.0005 | −0.0008, −0.0002 |
| Population density | −0.001 | −0.003, 0.001 | 0.007 | 0.004, 0.009 |

DOI: https://doi.org/10.7554/eLife.45474.012

**Table 3.** Association between environmental variables and the log incidence of dengue (over the last 5 years) for 211 municipalities in Colombia.

| Variable | Unadjusted | | Adjusted* | |
|---|---|---|---|---|
| | Estimate | 95% CI | Estimate | 95% CI |
| Elevation (per 100 m) | −0.02 | −0.03, 0 | 0.01 | −0.01, 0.03 |
| Mean temperature (per °C) | 0.01 | 0, 0.05 | | |
| Mean weekly precipitation | 0 | 0, 0 | −0.002 | −0.01, 0 |
| Population density | −0.04 | −0.07,−0.01 | −0.06 | −0.10, −0.02 |

* R2 of adjusted model: 0.04

DOI: https://doi.org/10.7554/eLife.45474.013

age 9 years for each of the subnational units represented in our data. Our results suggest that only a small minority of locations in Colombia and Brazil have > 80% seropositivity in this age group (*Figure 3*, *Figure 3—source data 1*). The expected proportion seropositive at 9 years of age ranged between 0.35 and 0.75 in provinces of Thailand; 0.13 and 0.85 in departments of Colombia; 0 and 0.88 in states of Brazil; and 0.07 and 0.56 in states of Mexico. The seroprevalence was estimated to be high enough in only 2/28 Colombian departments, 4/25 Brazilian states and none of the 72 Thai

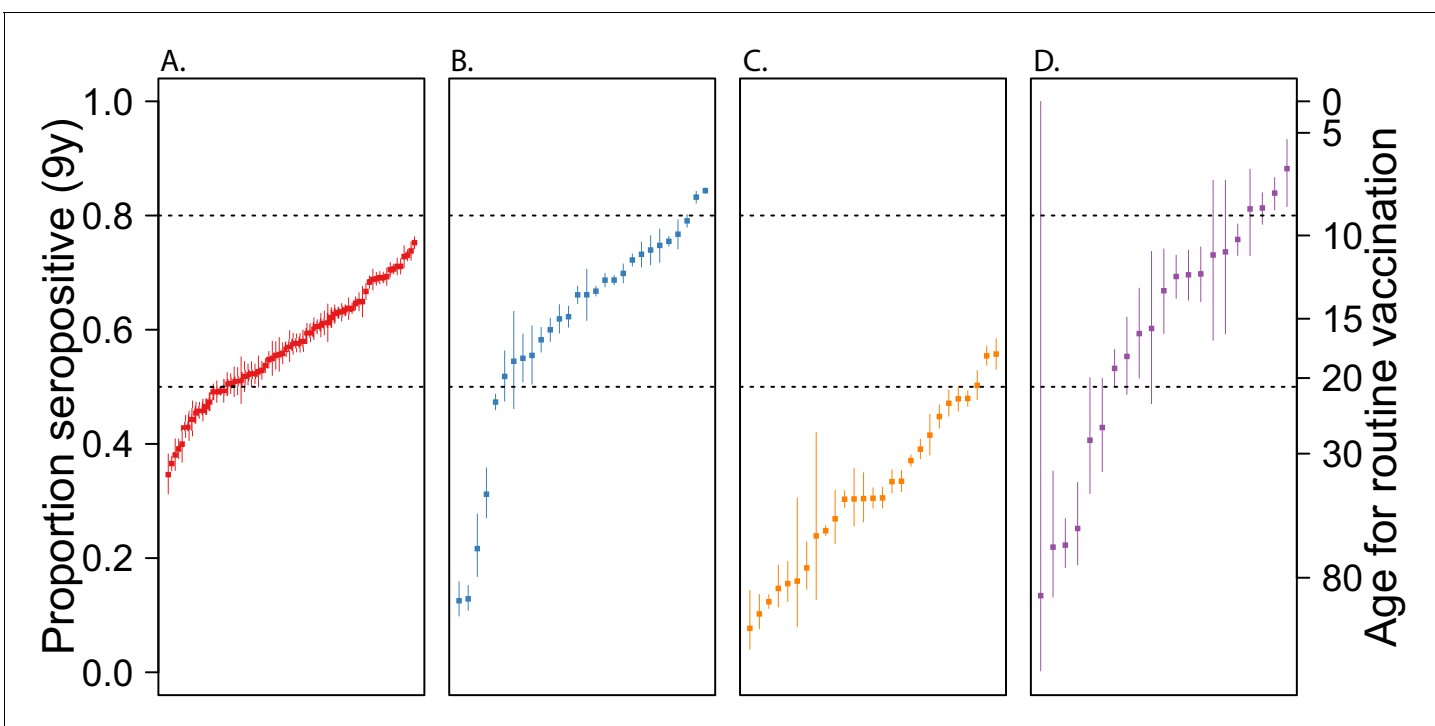

**Figure 3.** Guiding vaccination policy. Estimated dengue seroprevalence at 9 years of for administrative level 1 units of Thailand, Colombia, Brazil and Mexico Expected seroprevalence of dengue among children aged 9 years, derived from the FOI estimates (see Materials and methods for more details), for administrative level 1 units of Thailand (**A**), Colombia (**B**), Brazil (**C**) and Mexico (**D**). For each country, administrative units were ranked by their FOI. Dashed lines indicate 50% and 80% seroprevalence levels. Therefore, units above the 80% line are those where, according to the WHO-SAGE recommendation from 2018, it might be reasonable target children aged 9 years old for vaccination. Units below the 50% line are those where vaccination of this age-group would not be recommended. The axis on the right of the plot indicates the minimum age-group that would need to be targeted in each location to ensure at least 80% seropositivity.

DOI: https://doi.org/10.7554/eLife.45474.008

The following source data is available for figure 3:

**Source data 1.** Estimated forces of infection and seroprevalences for 148 spatial units.

DOI: https://doi.org/10.7554/eLife.45474.009

provinces or 27 Mexican states. Furthermore, even within the two Colombian departments that met the 80% seroprevalence threshold, only 9/13 (70%) of the municipalities evaluated reached this level. This proportion would probably be much lower had we considered all the municipalities in these departments, as those excluded had, on average, lower FOI (*Figure 2—figure supplement 2*).

In locations where seroprevalence among 9 year olds was estimated to be less than 80%, we calculated the age-group that could be targeted to ensure a seroprevalence > 80%. Our results suggest that, to comply with the WHO-SAGE recommendations, it would be necessary to target children 14 years of age or older in over 70% (108/148) of locations. Furthermore, in approximately 50% of the locations evaluated, it would be necessary to target individuals aged 18 years or older, precluding school-based vaccination strategies.

## Alternative metrics

Both the incidence and the mean adjusted incidence performed very poorly in ranking spatial units compared to the FOI (*Figure 1—figure supplement 1*). In contrast, the proportion of incidence under 10 years and the mean age of cases (to a lesser degree) performed quite well at ranking spatial units within countries. However, none of these metrics performed as well as the FOI when comparing spatial units from different countries, and they did not correlate as well with the gold standard (FOI estimates derived from seroprevalence data) (*Figure 1—figure supplement 2*).

## Sensitivity analyses

Since different surveillance systems have different reporting practices, we compared FOI estimates obtained when using data from severe cases alone (DHF or severe dengue) to estimates obtained when all dengue cases were considered jointly (*Figure 4*). This comparison was limited to Colombia, Brazil and Mexico because we only had DHF data from Thailand.

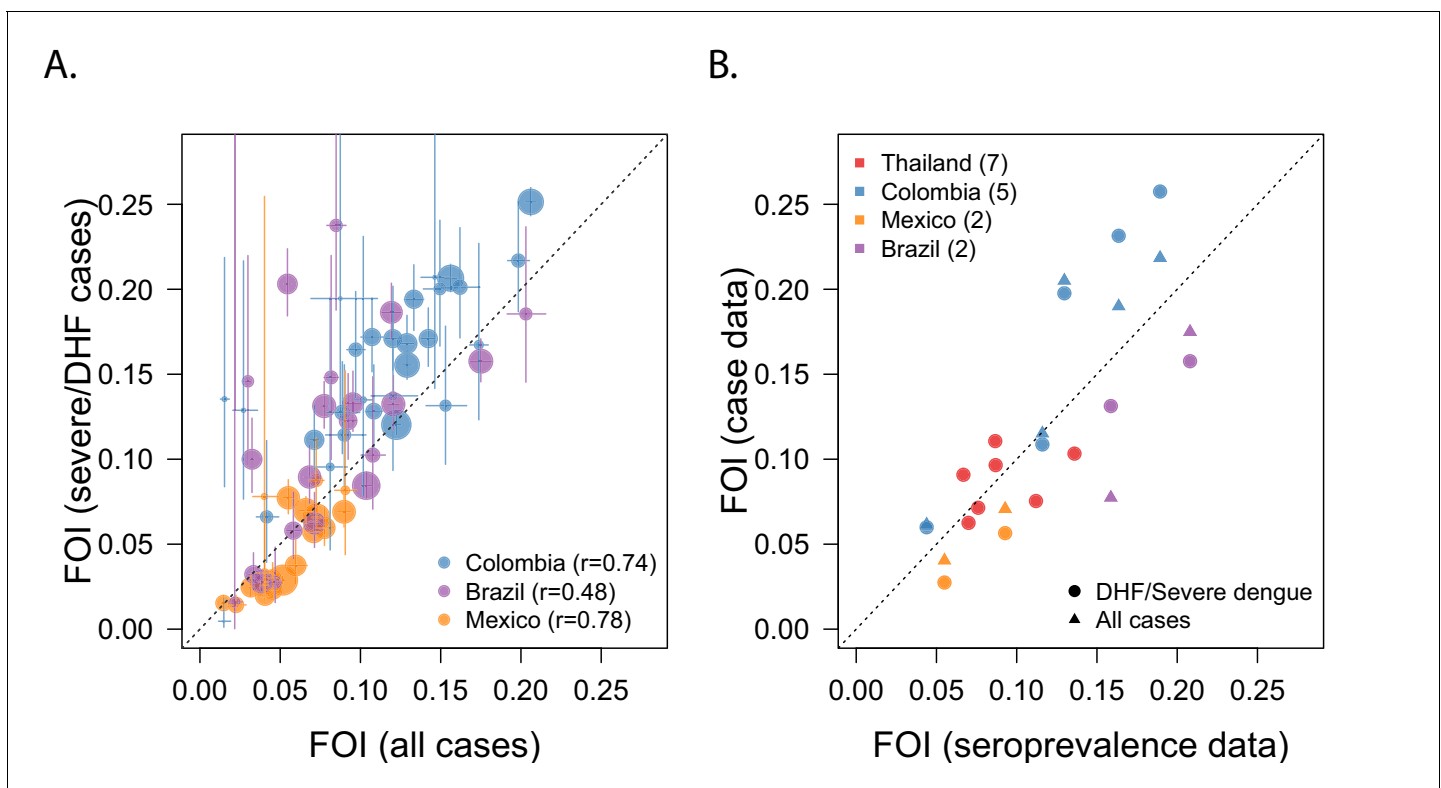

**Figure 4.** Sensitivity analysis: Comparing different types of data. (A) Correlation between estimates of the FOI obtained when using data from DHF/ severe dengue infections, vs. data from all cases combined. (B) Correlation between FOI estimates derived from different types of dengue data, with estimates derived from age-stratified serological data (gold standard).
DOI: https://doi.org/10.7554/eLife.45474.010

In general, estimates derived from all cases tended to underestimate those obtained from DHF cases, but the correlation between the estimates was good (*Figure 4A*) as well as the correlation with the gold standard (*Figure 4B*). The correlation was highest for Colombia and Mexico (r = 0.74 and r = 0.78, respectively), but less so for Brazil (r = 0.48). However, 2/3 of the Brazilian locations that showed discrepant results had the smallest number of DHF cases (n = 217 and 265) suggesting that sample size may have played a role. Similarly, all the Colombian locations where there was a large discrepancy had very low counts of DHF cases (n = 41, 55, and 48, respectively).

## Discussion

In this paper, we use dengue virus as an example to illustrate how age-specific incidence data can be used to quantify disease transmission and inform control interventions. We present a method to estimate the FOI of dengue from age-specific incidence data, and use it to generate country-wide subnational estimates for Thailand, Colombia, Brazil and Mexico. Our estimates correlate well with those derived from seroprevalence data (the gold standard), capture the expected spatial heterogeneity in risk, and outperform all other metrics traditionally used by surveillance systems, such as the crude incidence.

Most surveillance systems use crude case counts or incidences to describe temporal and spatial trends of communicable diseases. Our results underscore the extent to which, for immunizing infections in endemic circulation, recent incidence may be a poor metric of transmission and may be misleading when ranking spatial units (*Figure 1—figure supplement 2*). Immunity of the population in high-transmission settings reduces the number of individuals that are susceptible to infection. As a result, incidence in places where transmission intensity is lower, but people remain susceptible for a longer period of time, may be roughly equivalent to that in higher transmission intensity areas. In contrast, metrics such as the FOI, that quantify the risk among the susceptible population, better reflect the underlying transmission potential. Since FOI estimates are derived from the age distribution of incidence, and not from the aggregate counts, they are more robust to differences in surveillance efficiency and can be obtained with relatively small numbers of yearly cases.

Our estimates of the average transmission intensity of dengue in Thailand, Colombia, Brazil and Mexico are consistent with large variation in dengue transmission between and within countries. Spatial heterogeneity was substantial not only between, but also within administrative level one units. While this heterogeneity is probably driven by multiple environmental, socio-economic and demographic factors, our results suggest that as much as 35% of the variance may be explained by differences in temperature/elevation and population density alone. Transmission intensity was highest in northeastern Brazil, northern Colombia and eastern Thailand. Northeastern Brazil was also the region that experienced the highest incidence of Zika during the 2015/2016 epidemic, and this is not surprising given that both viruses are transmitted by the same mosquito vector (*A. aegypti*) (*Lowe et al., 2018*).

While age-stratified serological data remains the gold standard to quantify dengue transmission, our results illustrate how inferences derived from age-specific surveillance data could be used to inform control interventions such as vaccination. For example, the high transmission intensities (and expected seroprevalences) found in Northeastern Brazil suggest that this region might be ideal to implement pre-vaccination screening strategies. In addition, the large heterogeneity estimated in Colombia suggests that decisions of where to deploy control interventions, including vaccines, should be made at least at the municipal/district (administrative level 2) level. Since age-specific counts are routinely collected by many surveillance systems, they represent a unique opportunity to further our understanding of dengue burden and risk at spatial scales at which serological data is rarely available. While here we focus on presenting average forces of infection over 20 years, the same methods can also be used to estimate yearly forces of infection (*Hoang Quoc et al., 2016*; *Cummings et al., 2009*).

There are several limitations of using age-specific surveillance data to estimate transmission parameters of dengue. Of concern are differences in case definitions and reporting practices between and within countries. While the case-definition of DHF is quite specific, milder forms of the disease are characterized by a non-specific febrile syndrome. Therefore, data from locations that report 'all cases' rather than more severe forms (DHF) are more likely to be misclassified, and could be less useful to infer transmission histories. Nevertheless, our results and sensitivity analyses are

encouraging and suggest that that even data from all dengue cases can be useful to infer transmission patterns.

Our model also makes several assumptions that may be questionable. It assumes that the age distribution of cases represents the age distribution of secondary infections, thus ignoring the potential contribution of primary, tertiary and quaternary infections. It also assumes that risk of infection, symptoms, and health seeking behavior are not age dependent, even though there is some evidence that suggests that this may not be the case(*Katzelnick et al., 2018*; *Guzmán and Kourí, 2002*; *Thai et al., 2011*). It assumes equal circulation of all serotypes, despite the known dominance of specific serotypes for extended periods of time in the Americas (*Katzelnick et al., 2018*; *Teixeira et al., 2009*). Finally, it estimates transmission intensities over extended periods of time (20 years), averaging over variations in the FOI that may occur at shorter time scales. Despite these simplifications, validation of our estimates using age-stratified serological data from 16 locations is very encouraging, as is the good correlation with known drivers of dengue transmission. While further validation is desirable, it is important to note that some discrepancy between our estimates and those derived from seroprevalence data is expected as the two sources of data do not represent exactly the same period of time and location. For example, most of the serosurveys available were conducted in specific urban centers, while case data represents the full administrative unit.

Targeting control interventions against dengue and other communicable diseases requires good understanding of when and where transmission is occurring. Careful analyses of age-specific incidence data can provide very useful information to characterize transmission across time and space. While here we focus on dengue, the same approach should be generalizable to other immunizing infections including chikungunya and Zika. Open access to age-specific incidence data would greatly enrich and enhance existing efforts to quantify trends in the global burden of disease.

## Materials and methods

### Data used

We used data on the yearly age-specific reported number of dengue cases for administrative level 1 units of Thailand, Colombia, Brazil and Mexico (*Ministério da Saude, 2017*; *Bureau of Epidemiology, Department of Disease Control, 2019*; *Direccion General de Epidemiología, 2017*; *Instituto Nacional de Salud Colombia, 2016*) as well as administrative level two units from Colombia. The period of time available varied between countries, but ranged between 5 years in Colombia and 25 years in Thailand. Where possible (Thailand and Brazil) we used age-specific case reports on dengue hemorrhagic fever (DHF) for our analyses. However, for both Colombia and Mexico, we had to use data on all reported dengue cases because the number of DHF cases reported by a substantial number of spatial units (9/32 units in Colombia and 10/31 in mexico) were insufficient to estimate the FOI. We also used population data from each administrative unit analyzed, available from the national statistical office of each country. Information on the type (e.g. DHF vs. all dengue), source and years of data used are provided in *Table 1*.

### Statistical analyses

#### Estimating the force of infection

We estimated the average force of infection (FOI, $\lambda$) of dengue, over the last 20 years, for each administrative unit for which we had available data. The FOI is a metric used to characterize the transmission intensity in a specific setting and estimates the per capita rate at which susceptible individuals are infected. Methods to estimate transmission intensity from age-specific incidence data have been previously used to reconstruct the transmission history of measles and dengue (*Grenfell and Anderson, 1985*; *Cummings et al., 2009*). Briefly, these methods rely on the fact that, for immunizing infections, accumulation of immunity shapes the age distribution of future cases. In settings with high endemic transmission, incident cases are expected to be concentrated in younger age groups, as adults are likely to be already immune (*Figure 1A*). In contrast, in places where there is less population immunity, the age distribution of cases is more likely to resemble the age distribution of the population itself, with cases in in both children and adult populations.

Methods to estimate dengue forces of infection from case data have been applied to settings where dengue is thought to be close to endemic circulation (*Cummings et al., 2009*; *Imai et al.,*

*2016*). These methods generally rely on the cumulative incidence proportion, and therefore assume that all individuals are infected by dengue at some point in their lifetime. They also often assume that the distribution of cases (of dengue hemorrhagic fever cases in particular) is representative of the distribution of secondary cases. Here, we extend these methods to accommodate settings where transmission hazards are lower or where dengue may have been more recently introduced. We do this by modeling directly the age-specific incidence of cases, rather than the cumulative incidence proportion.

The fraction of the population susceptible to all dengue serotypes at age a and t, x(a,t) is given by,

$$x(a,t) = e^{\int_0^a -4\lambda(t-\tau)d\tau}. \tag{1}$$

where $\lambda(t)$ is the average FOI per serotype at time t and $4\lambda(t)$ is the total FOI assuming four circulating serotypes. The proportion of individuals of age a who have been infected with only serotype at time t, but are still susceptible to all other serotypes is denoted z1(a,t) and is given by:

$$z_1(a,t) = 4\left(e^{-\int_0^a 3\lambda(t-\tau)d\tau}\right)\left(1 - e^{-\int_0^a \lambda(t-\tau)d\tau}\right). \tag{2}$$

Assuming that the age-specific incidence of cases is representative of the distribution of secondary infections, the expected incidence rate among individuals age a at time t is given by

$$I(a,t) = 3\lambda(t)z_1(a,t) \tag{3}$$

and the expected reported number of cases is

$$(a,t) = (I(a,t)P(a,t)\phi(t)), \tag{4}$$

where P(a,t) is the size of the population aged a at time t, and $\Phi(t)$ represents a time specific reporting rate/scaling factor.

## Likelihood and estimation

Assuming that the observed age-specific case counts C(a,t) follow a Poisson distribution, the likelihood of the data can be expressed as

$$L(C|\lambda,\phi) = \prod_t\prod_a \frac{\left(\Lambda(a,t)^{C(a,t)}e^{-\Lambda(a,t)}\right)}{C(a,t)!}, \tag{5}$$

We fit the model in a Bayesian Markov chain Monte Carlo (MCMC) framework using the RStan package in R (*Stan development Team, 2019*) (*The R Foundation, 2019*). Both the annual hazards of infection ($\lambda$) and the reporting rates ($\Phi$) were estimated on a logit scale using wide priors (Normal distribution with mean 0 and standard deviation of 1000). We simulated four independent chains, each of 30000 iterations and discarded the firs 10000 iterations as warm-up. We assessed convergence visually and using Rubin's R statistic. We obtained 95% credible intervals from the 2.5% and 97.5% percentiles of the posterior distributions. Code to implement the model is available at https://github.com/isabelrodbar/dengue_foi. (*Rodriguez-Barraquer, 2019*; copy archived at https://github.com/elifesciences-publications/dengue_foi).

A limitation of this approach is that, due to the large number (often in the thousands) that are characteristic of the data for some settings, the estimated confidence intervals produced are extremely narrow and do not reflect the underlying uncertainty adequately. The observed counts can also be assumed to follow a negative binomial distribution to account for some overdispersion.

## Parameters estimated

Since it is known that the FOI has varied substantially over time in many of the settings considered, we allowed $\lambda(t)$ to vary as a function of time. To limit the number of parameters estimated, we assumed constant $\lambda(t)$ for periods of 20 years. Thus, if for a given setting we were estimating hazards for the period 1935–2015, we assumed piecewise-constant $\lambda(t)$s for the periods 1935–1954, 1955–1974, 1975–1994, 1995–2015. Given the objective of this study was to characterize recent

transmission in endemic settings, we focused our results on the estimate of the average λ(t) for the most recent 20-year period but present other estimates in the supplementary material (*Appendix 1-Figure 1*). Rather than reporting λ(t), we focus on reporting the total FOI (4λ(t)).

### Estimating the proportion expected to be seropositive at a given age

Using our estimates of the average FOI, we estimated the proportion of individuals expected to be seropositive by age y(x) as:

$$\begin{aligned} y(x) &= 1 - x(a) \\ &= 1 - e^{-4\lambda a} \end{aligned}$$

where x(a) is the proportion of the population susceptible at age a and λ is the average FOI per serotype (assuming four serotypes circulating). Since the vaccine has been registered for use in children 9 years of age or older, we report the proportion of individuals expected to be seropositive by age 9 years for each of the settings.

### Estimating the minimum age to achieve a given level of seropositivity

Given that the WHO recommended using this vaccine in places where at least 80% of the target age group is seropositive, we estimated the minimum age at which this level of seropositivity is expected for each of the settings. For a given level of transmission λ it is possible to estimate the minimum age (A) at which a given level of seropositivity (s) is expected as:

$$A(s) = \frac{-\log(1-s)}{4\lambda}$$

## Validation and sensitivity analyses

We validated our estimates of the FOI by comparing them to estimates obtained from age-stratified serological data (the gold standard) for 16 locations for which both serologic and age-specific case data was available. Methods to estimate forces of infection from seroprevalence data have been previously described (*Ferguson et al., 1999*; *Rodriguez-Barraquer et al., 2011*).

Since dengue transmission is known to be highly spatially heterogeneous, we also correlated our administrative level two estimates for Colombia with known environmental drivers of dengue transmission: temperature, elevation, population density and a published composite metric of *A. aegypti* abundance (*Siraj et al., 2018*).

As stated above, a key assumption of this model is that the age distribution of cases represents the age distribution of secondary infections. Data from Thailand has consistently suggested that the majority of dengue hemorrhagic fever (DHF) cases arise from secondary infections (*Burke et al., 1988*) and therefore we limited our analysis to reports of DHF where possible (Thailand, Brazil). However, for Colombia and Mexico we used data from severe and non severe cases because the severe dengue data alone was too sparse. To assess the impact of the data type, we compared estimates obtained from DHF/severe cases alone, to those obtained when all dengue cases were considered.

## Alternative metrics

Since estimating the FOI requires fitting parametric models to the data, we explored whether alternative summary metrics, computed directly from the age-specific case data, could be equally useful in ranking spatial units within and across countries. The metrics considered included some that are commonly used by surveillance systems such as a) crude incidence and b) standardized mean incidence, but also alternative metrics that are not commonly used such as c) mean age of cases, d) cumulative proportion of incidence occurring by age 10 years and e) age at peak incidence. For each spatial unit, we computed these metrics using the most recent 5 years of data available.

## Application: guiding dengue vaccination policy

The first dengue vaccine has been licensed for use in children over 9 years of age in 20 countries. Due to uncertainty regarding the vaccine's benefits and risks in individuals who haven't been previously infected by dengue, the WHO's scientific advisory group of experts (SAGE) committee recommended in April 2016 that this vaccine only be used in settings with known high endemicity, defined

as places where seroprevalence is greater than 70% in the target vaccination age-group (*SAGE, 2016*), and should not be used in places where seroprevalence is under 50%.

This recommendation was later revised, and the WHO now recommends that individuals should be tested for dengue antibodies prior to vaccination, and the vaccine should only be given to individuals who have been infected by dengue in the past (*WHO, 2018*). In the absence of appropriate serological assays that would allow for pre-vaccination screening, an alternative that has been discussed is deploying the vaccine in settings were seroprevalence is 80% or greater. These recommendations pose challenges to countries wanting to implement the vaccine, as they require detailed knowledge of the epidemiology of dengue. Specifically, they require knowledge of the population seroprevalence against dengue at subnational levels, even though such data is not available.

In order to provide information useful to countries considering deploying the vaccine according to the WHO recommendations, we used our estimates of the FOI to calculate the proportion of the population expected to be seropositive at age 9 years for of the subnational units represented in our data. We also estimated the minimum age group expected to have a seroprevalence of 80% or greater. In collaboration with the MRC center for Outbreak Analysis and Modeling, these estimates were made available online in June 2017 (https://mrcdata.dide.ic.ac.uk/_dengue/dengue.php) as a tool to help countries deciding where to target vaccination.

## Acknowledgements

We thank the Ministry of Public Health (Thailand), the Instituto Nacional de Salud (INS, Colombia), the Ministerio da Saude (Brazil) and the Direccion General de Epidemiología. Sistema Nacional de Vigilancia Epidemiológica (Sinave, Mexico), for making the data necessary for these analyses publicly available. We also thank Sompong Vongpunsawad for sharing the raw data of the serological studies conducted in Thailand. We thank Neil Ferguson, Natsuko Imai and Wes Hinsley for valuable input and for making these estimates publicly available through the Global Dengue Transmission Map website.

## Additional information

### Funding

| Funder | Grant reference number | Author |
|---|---|---|
| National Institutes of Health | R01AI114703-01 | Derek A Cummings |
| European Research Council | 804744 | Henrik Salje |

The funders had no role in study design, data collection and interpretation, or the decision to submit the work for publication.

### Author contributions

Isabel Rodriguez-Barraquer, Conceptualization, Data curation, Formal analysis, Investigation, Visualization, Methodology, Writing—original draft, Project administration, Writing—review and editing; Henrik Salje, Formal analysis, Writing—original draft, Writing—review and editing; Derek A Cummings, Conceptualization, Data curation, Formal analysis, Writing—original draft, Writing—review and editing

### Author ORCIDs

Isabel Rodriguez-Barraquer (iD) https://orcid.org/0000-0001-6784-1021
Henrik Salje (iD) https://orcid.org/0000-0003-3626-4254

### Decision letter and Author response

Decision letter https://doi.org/10.7554/eLife.45474.031
Author response https://doi.org/10.7554/eLife.45474.032

## Additional files

### Supplementary files
• Transparent reporting form
DOI: https://doi.org/10.7554/eLife.45474.014

### Data availability
The code to implement the model described in our study is available at https://github.com/isabel-rodbar/dengue_foi (copy archived at https://github.com/elifesciences-publications/dengue_foi). The case data used for the analyses is publicly available and can be accessed through the following links links: Brazil- http://tabnet.datasus.gov.br/cgi/deftohtm.exe?sih/cnv/mruf.def; Thailand - http://www.boe.moph.go.th/boedb/surdata/index.php; Colombia - http://www.ins.gov.co/lineas-de-accion/Sub-direccion-Vigilancia/sivigila/Paginas/vigilancia-rutinaria.aspx and https://www.sispro.gov.co/Pages/Home.aspx; Mexico - http://www.epidemiologia.salud.gob.mx/anuario/html/anuarios.html.

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

## Appendix 1

DOI: https://doi.org/10.7554/eLife.45474.015

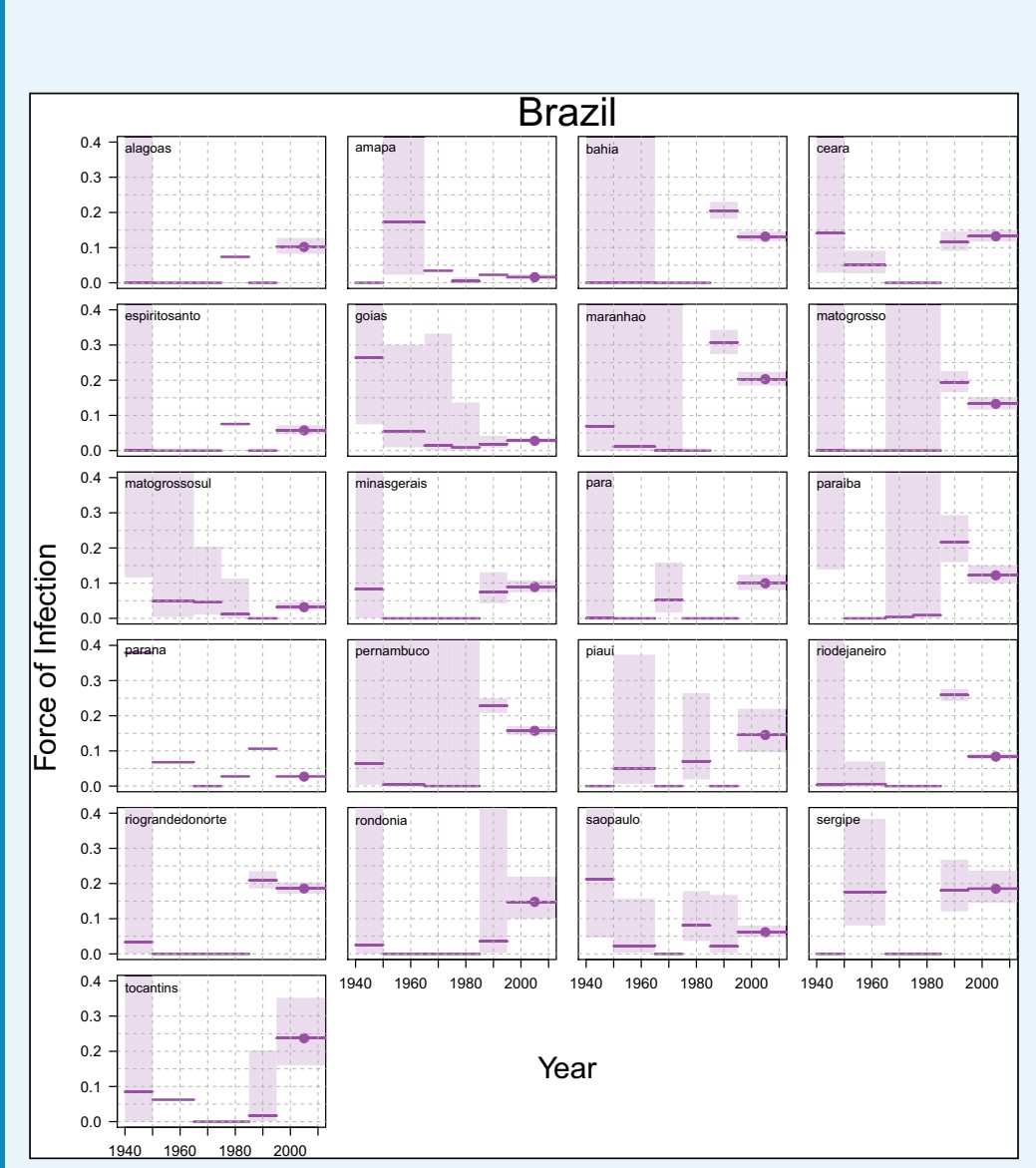

**Appendix 1—figure 1.** Time varying Force of Infection Estimates.

DOI: https://doi.org/10.7554/eLife.45474.016

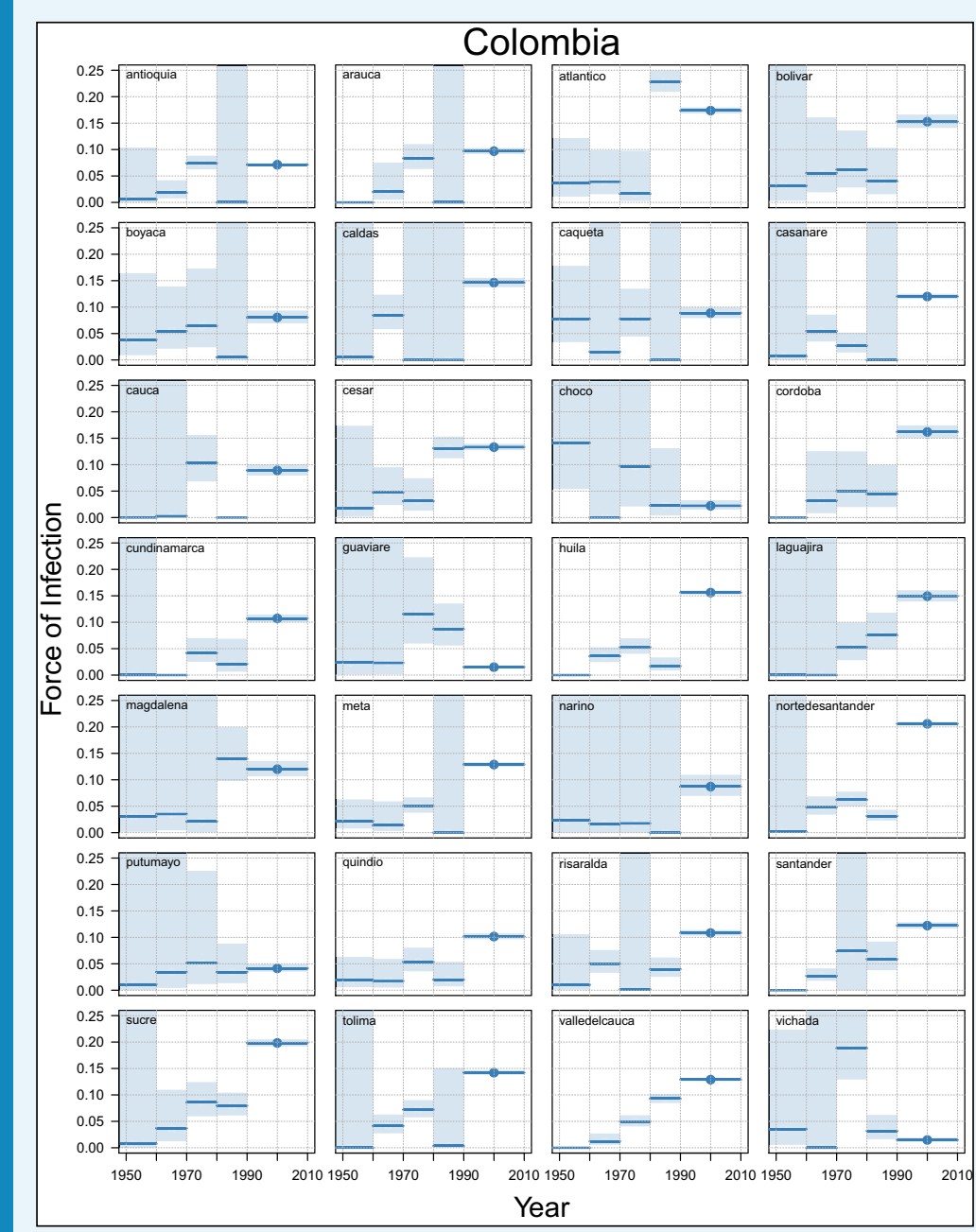

**Appendix 1—figure 2.** Time varying Force of Infection Estimates, Colombia.

DOI: https://doi.org/10.7554/eLife.45474.017

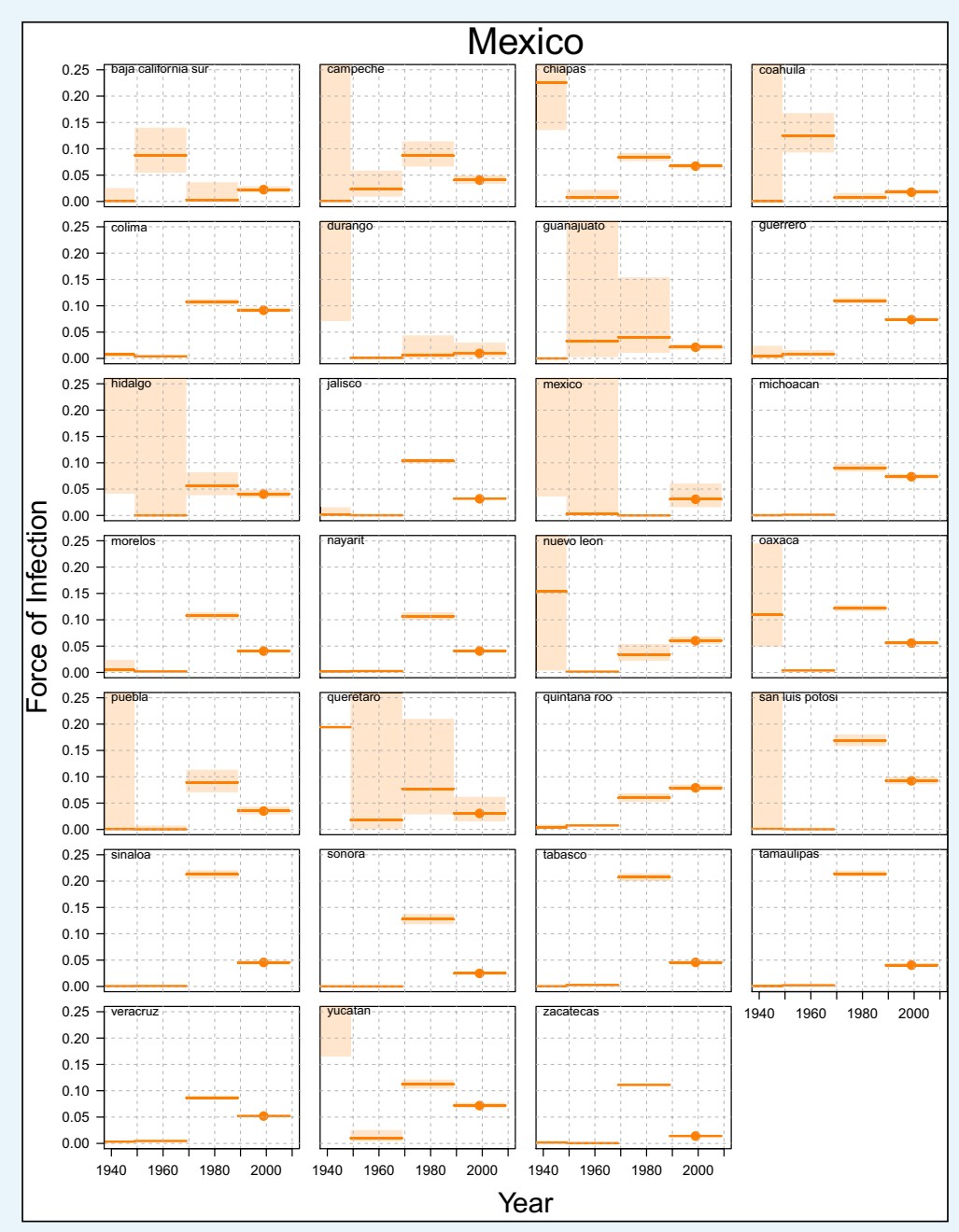

**Appendix 1—figure 3.** Time varying Force of Infection Estimates, Mexico.

DOI: https://doi.org/10.7554/eLife.45474.018

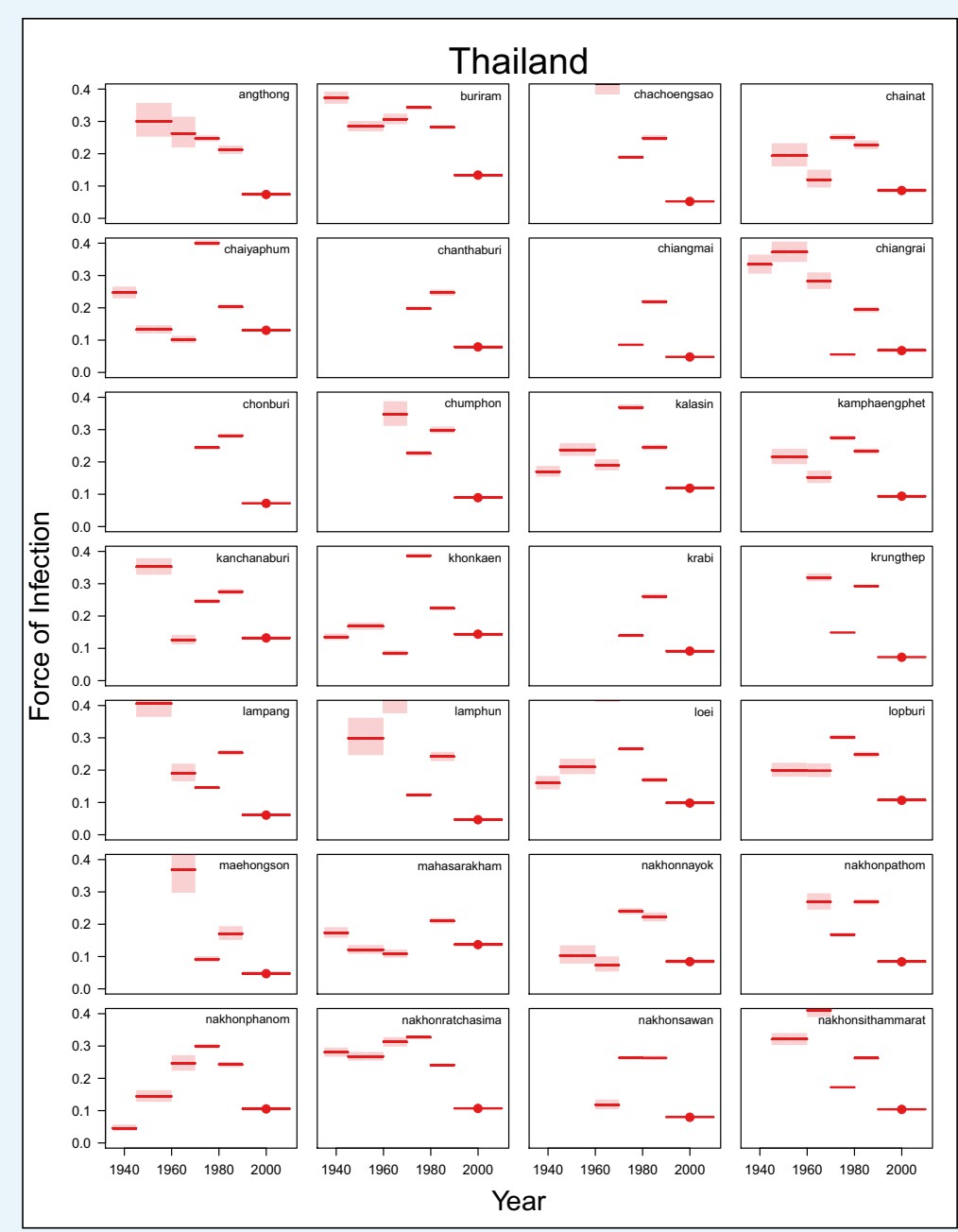

**Appendix 1—figure 4.** Time varying Force of Infection Estimates, Thailand (1).

DOI: https://doi.org/10.7554/eLife.45474.019

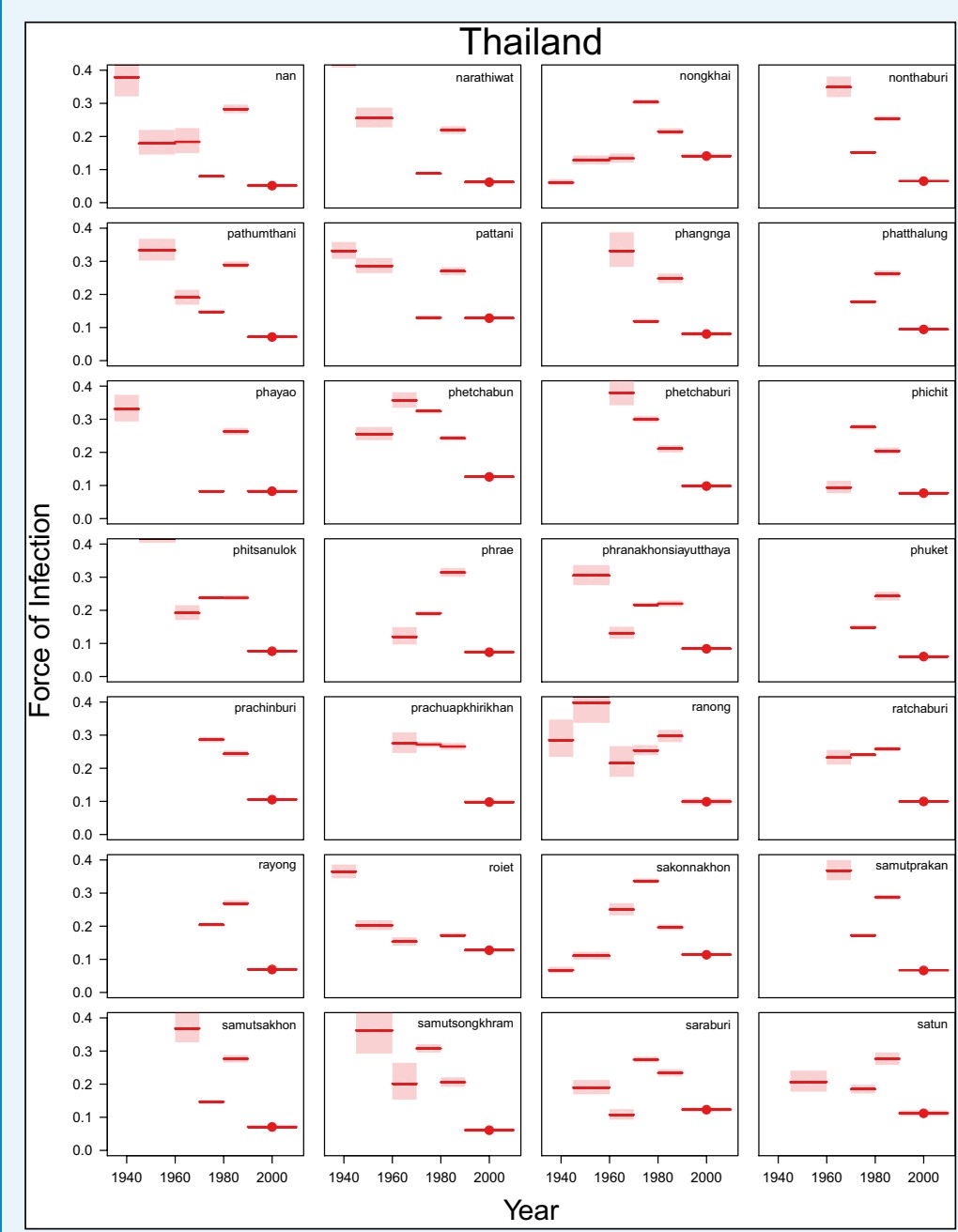

**Appendix 1—figure 5.** Time varying Force of Infection Estimates, Thailand (2).

DOI: https://doi.org/10.7554/eLife.45474.020

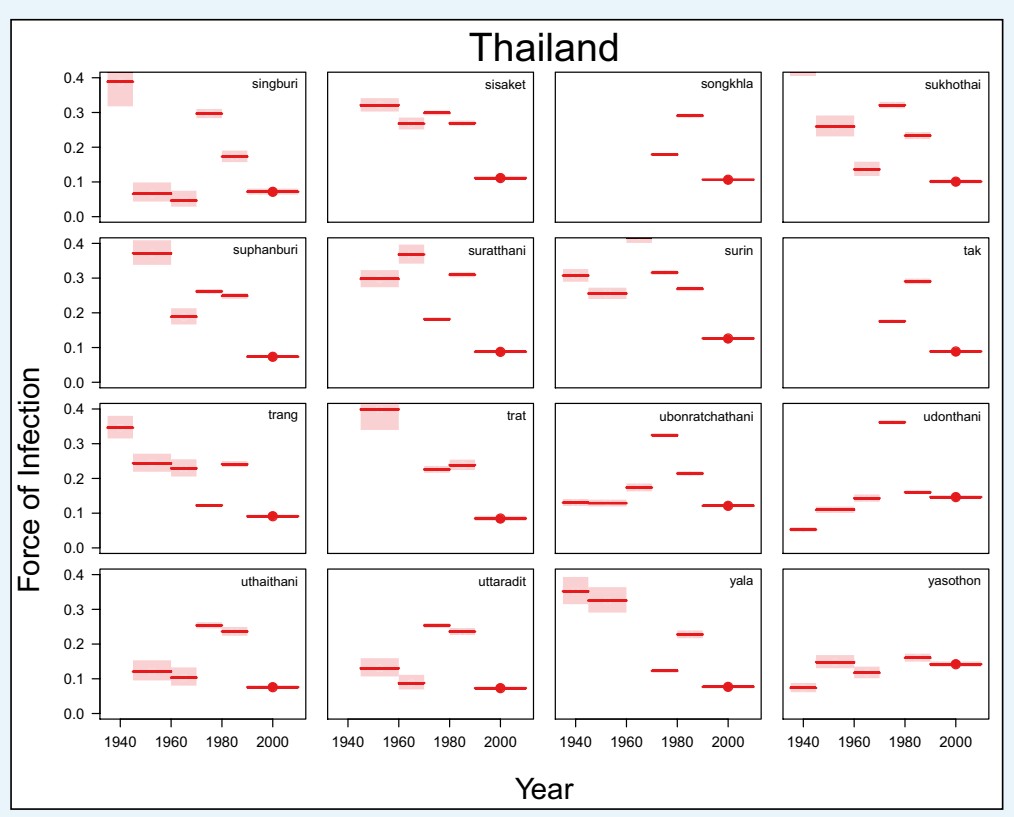

**Appendix 1—figure 6.** Time varying Force of Infection Estimates, Thailand (3).
DOI: https://doi.org/10.7554/eLife.45474.021

## Appendix 2

DOI: https://doi.org/10.7554/eLife.45474.015

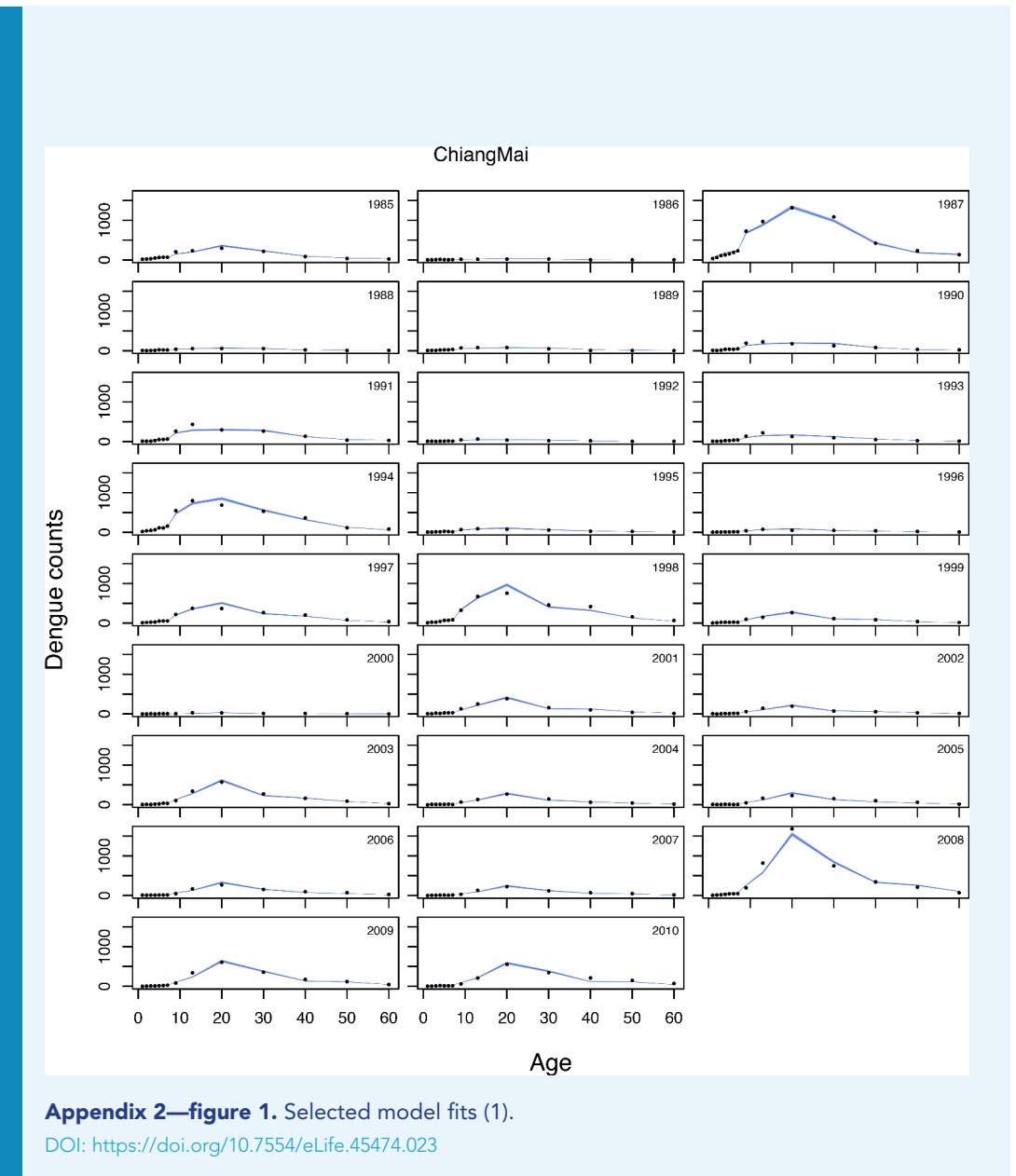

**Appendix 2—figure 1.** Selected model fits (1).

DOI: https://doi.org/10.7554/eLife.45474.023

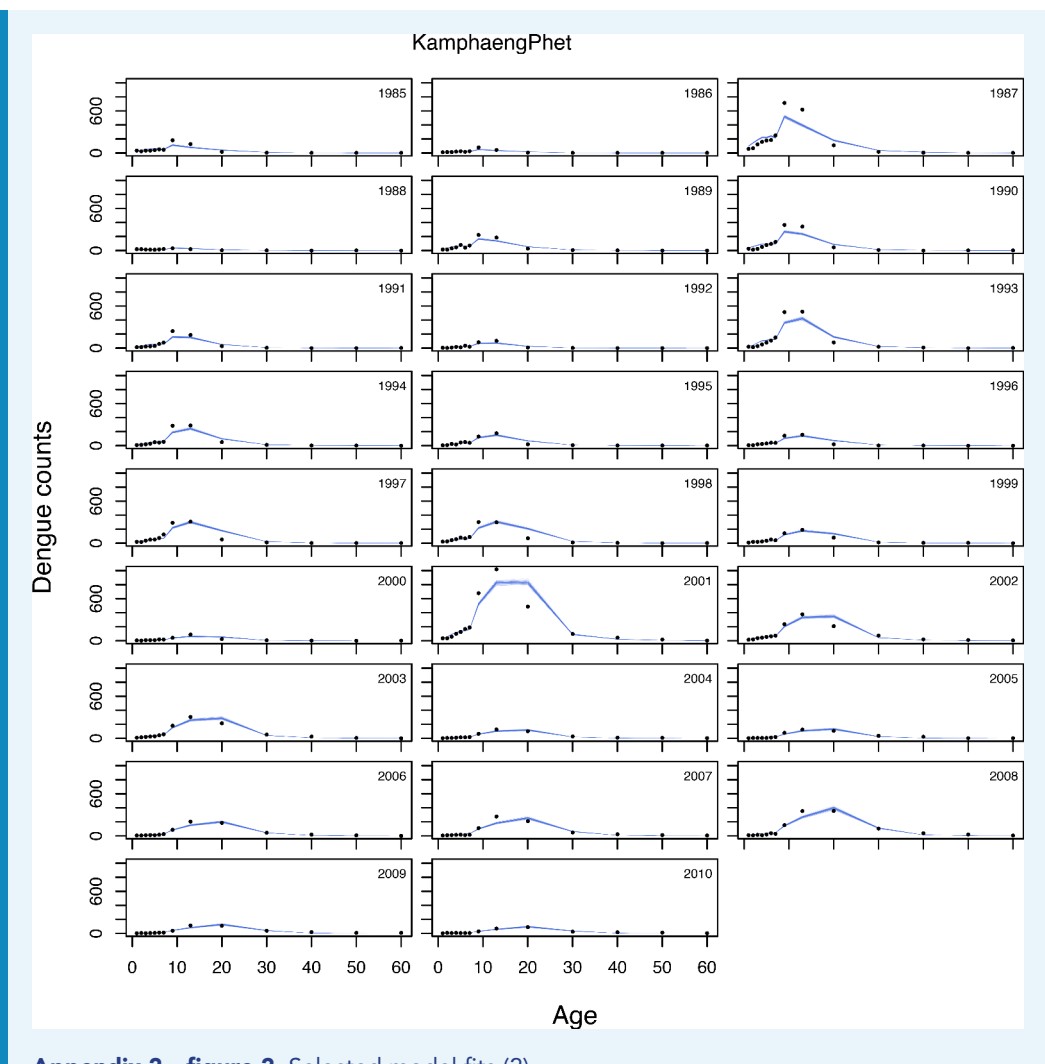

**Appendix 2—figure 2.** Selected model fits (2).

DOI: https://doi.org/10.7554/eLife.45474.024

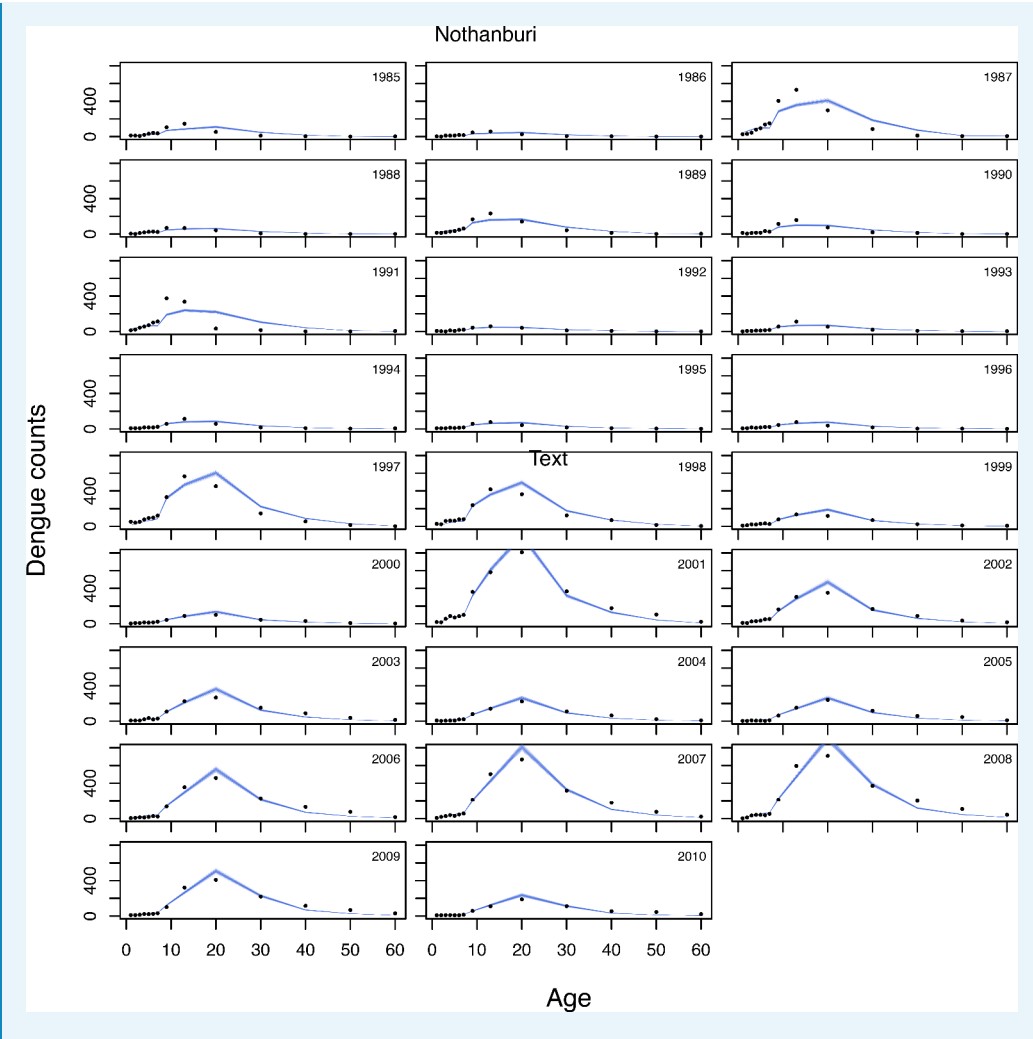

**Appendix 2—figure 3.** Selected model fits (3).

DOI: https://doi.org/10.7554/eLife.45474.025

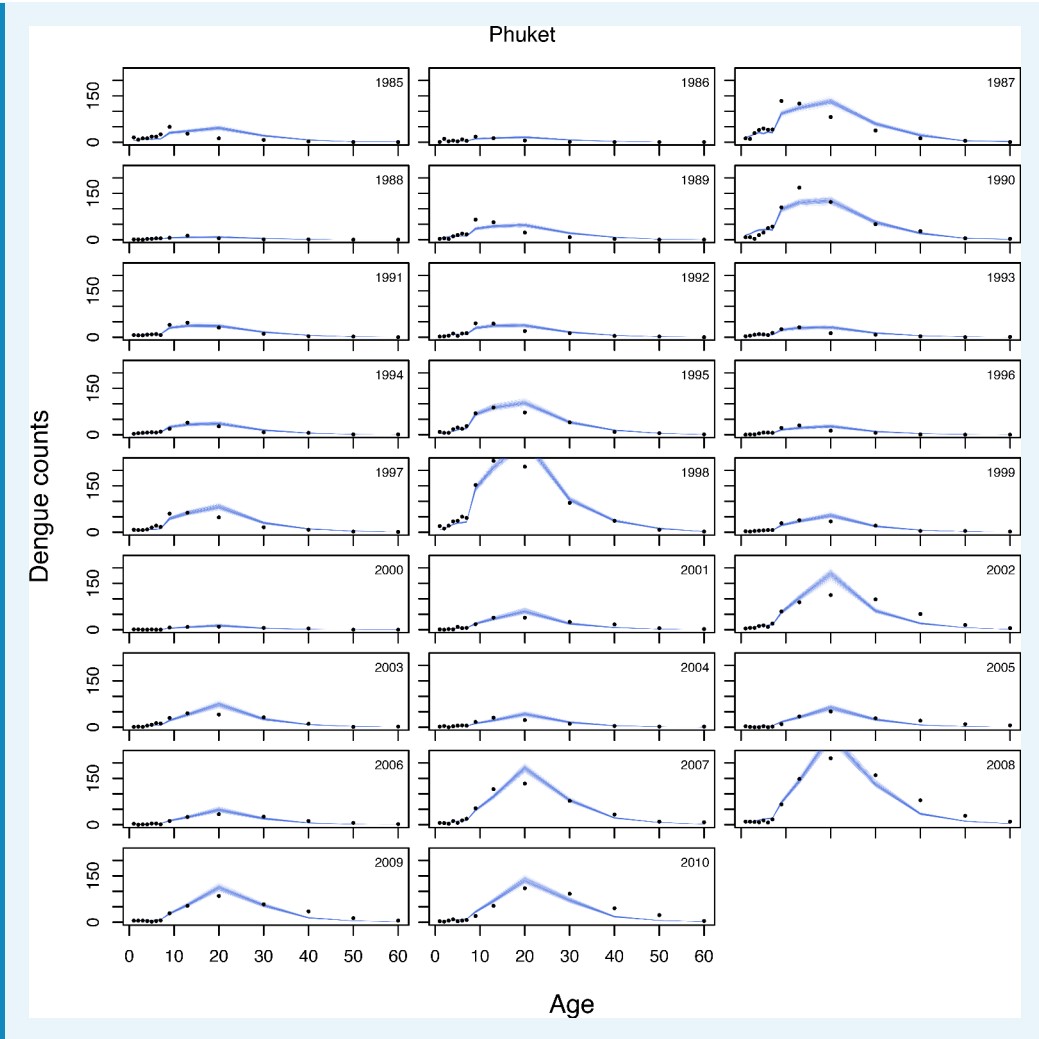

**Appendix 2—figure 4.** Selected model fits (4).

DOI: https://doi.org/10.7554/eLife.45474.026

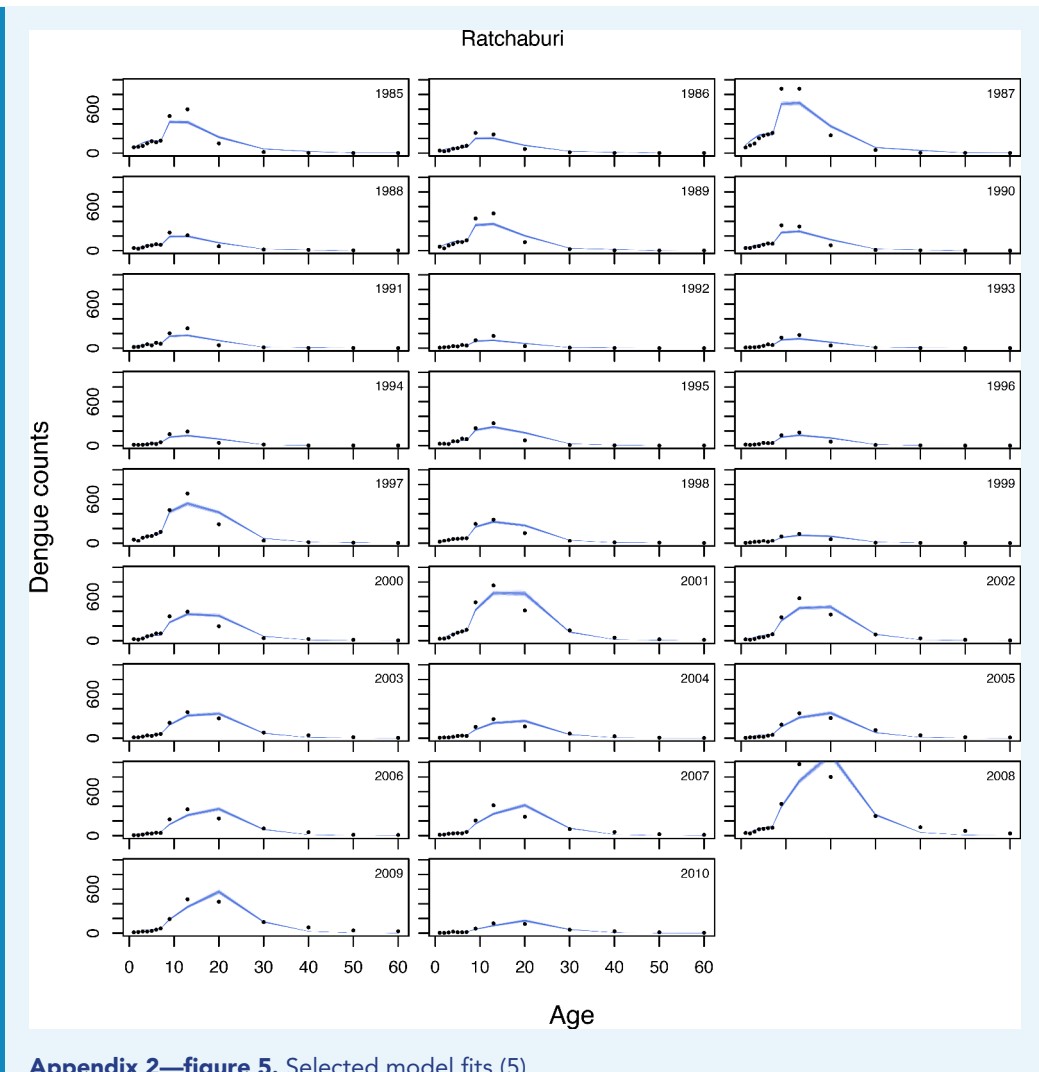

**Appendix 2—figure 5.** Selected model fits (5).

DOI: https://doi.org/10.7554/eLife.45474.027

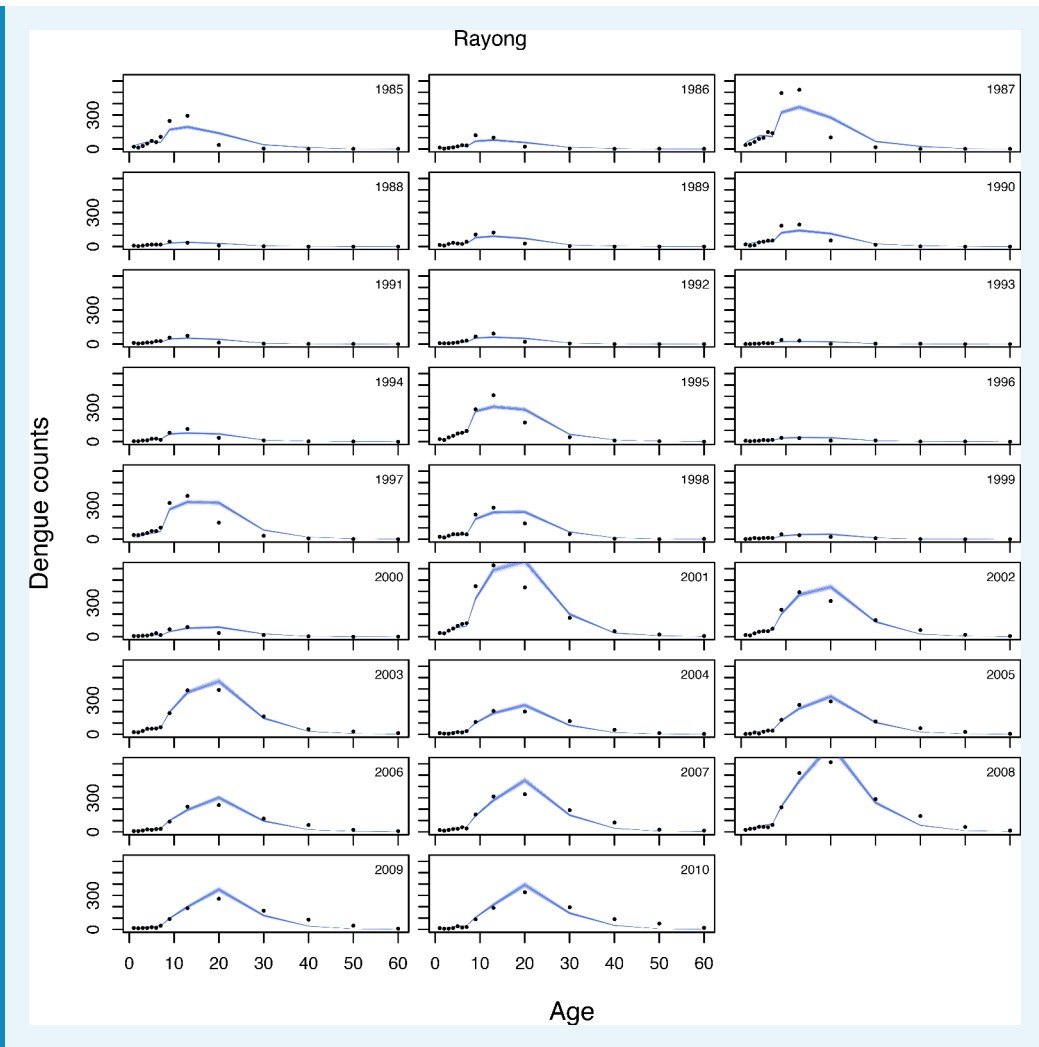

**Appendix 2—figure 6.** Selected model fits (6).

DOI: https://doi.org/10.7554/eLife.45474.028

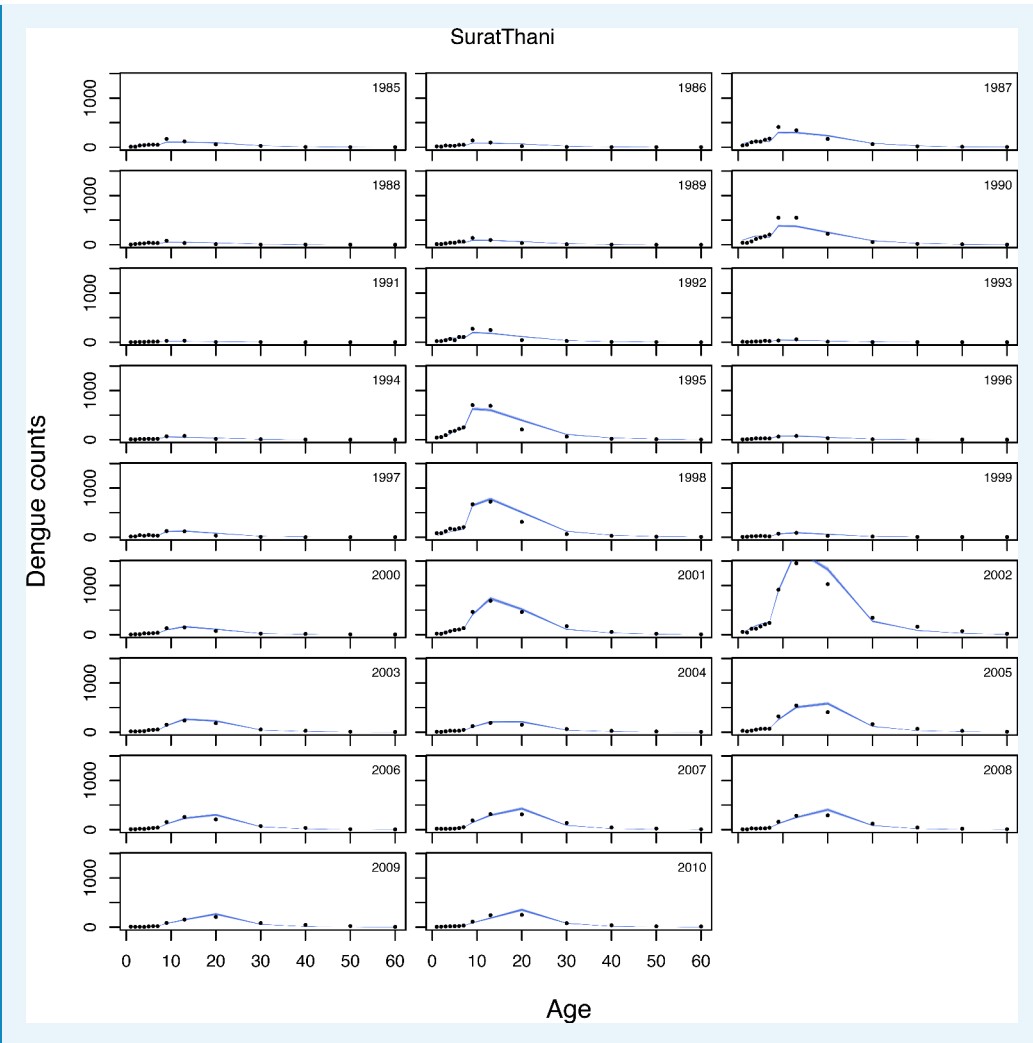

**Appendix 2—figure 7.** Selected model fits (7).

DOI: https://doi.org/10.7554/eLife.45474.029

