## [Decision Letter]

Thank you for submitting your article "Opportunities for improved surveillance and control of infectious diseases from age-specific case data" for consideration by *eLife*. Your article has been reviewed by three peer reviewers, including Jos WM van der Meer as the Reviewing Editor and Reviewer #1, and the evaluation has been overseen by Eduardo Franco as the Senior Editor. The following individual involved in review of your submission has agreed to reveal their identity: Oliver Brady (Reviewer #3).

The reviewers have discussed the reviews with one another and the Reviewing Editor has drafted this decision to help you prepare a revised submission.

Summary:

This interesting paper provides useful insight into the challenging problem of identifying populations suitable for the deployment of the dengue vaccine, given the risk of side effects in naïve individuals upon exposure to the virus. There are however several concerns and major questions about the assumptions and their implications on the relevance of the work in terms of programmatic recommendations. Key details on the methods are lacking, so additional details are needed as they are important to ensure the correct interpretation of the findings.

Essential revisions:

1) One of the justifications for their model is that it is better able to predict dengue infections than using recent incidence. This was used both in comparisons with the FOI in the regression analysis and also in the supplementary plots correlating ranks etc. However, 'recent incidence' is not defined. Was this in the most recent year of data? The average over the 20-year block? Are the results sensitive to this definition? How does an age-adjusted incidence metric compare to the FOI? If a simpler approach is 'good enough', would not this be more operationally tractable?

2) The authors correctly assume that all 4 serotypes could be co-endemic, and that DHF cases are likely to represent the second infection, with mild cases most likely a person's 1st-4th infection. The focus on the issues with the current vaccination recommendations is very relevant, but it is not very clear in the text whether the authors considered 'likely seropositive' after expected 1st infection (best candidate for vaccine) or whether it was any infection, with 2nd-4th not necessarily needing the vaccine as are already protected due to the homology between the strains. In epidemic-prone settings, this is somewhat simplified, but in areas with endemic transmission and co-circulating strains this is quite complex. If the model does in fact represent the likely first infection, then these nuances should be better described. If it regards the later infections, more justification for the relevance of the estimates should be included.

3) To validate the estimates, the authors have plotted FOI against age-stratified serological data. However, how was case data (as labelled on the axis) estimated from FOI? Also, how were the seroprevalence estimates generated from the serological data and does this represent seropositive to all/at least 1 etc. serotypes?

4) In addition to the limitations on the described method for calculating FOI, the paper would benefit from a few sentences in the Discussion on the limitations of using (very) long-term average FOI at a highly aggregate level to dictate control policy. Such measures will be insensitive to shorter-term changes (e.g., to interventions or a wide variety of factors that act at < 20-year intervals) and finer-scale heterogeneities clearly occur and clearly other metrics of transmission still have a role to play in deciding dengue control policy. Suggesting that these maps might be useful for prioritising the areas where more detailed serosurveys (or individual test and vaccinate) should be done in first instance would be a more conservative interpretation than stating that vaccinating in this particular province of Brazil is "highly questionable".

5) The assumptions and limitations of the described method are well detailed in the Discussion, but it would be interesting if there were any patterns in the residuals. Are there any circumstances where this method is better or worse?

6) A major advantage of this method is international comparability, yet (as acknowledged by the authors in the Materials and methods section, but not the Results and Discussion), it is affected by different case definitions of dengue and severe dengue used in each country. Supplementary Figure 4 shows some interesting discrepancies that are not mentioned in the Results section. Thus, how generalisable are the results in countries that have much less well-defined case definitions than the countries chosen here?

7) Regarding the methods: the use of additive force of infection (i.e. 3*λ) seems strange and the reviewers wonder whether it quite adds up due to the multiple permutations of ways in which a person can have a secondary infection. In a multi serotype model with equal FOI between serotypes, the probability of infection with any serotype is 1-(1-λ)^4, i.e., 1 minus probability of no infection to any serotype and slightly lower than 4*λ. This would be worth checking as it definitely affects secondary infection attack rate.

---

## [Author Response]

Essential revisions:1) One of the justifications for their model is that it is better able to predict dengue infections than using recent incidence. This was used both in comparisons with the FOI in the regression analysis and also in the supplementary plots correlating ranks etc. However, 'recent incidence' is not defined. Was this in the most recent year of data? The average over the 20-year block? Are the results sensitive to this definition? How does an age-adjusted incidence metric compare to the FOI? If a simpler approach is 'good enough', would not this be more operationally tractable?

We apologize for the lack of clarity in terms of the metric of incidence used. Since we wanted to compare our FOI estimate to a metric of recent incidence, but acknowledging that dengue is highly variable between years, we computed the average over the last five years of available data.

We agree with the reviewers that a simpler approach would be useful and this was something that we extensively discussed with the WHO. We have compared estimates to a number of metrics, including some that are commonly used by surveillance systems such as (a) crude incidence and (b) standardized mean incidence, but also alternative metrics that are not as commonly used such as (c) mean age of cases, (d) cumulative proportion of incidence occurring by age 10 years (Cum10) and (e) age at peak incidence. We computed these metrics for each spatial unit using the most recent 5 years of data available.

As shown in Figure 1—figure supplement 1, the incidence metrics perform very poorly in ranking spatial units compared to the FOI. In contrast, Cum10 and the mean age of cases (to a lesser degree), perform quite well at ranking spatial units within countries. However, none of the metrics performs as well as the FOI when comparing countries, and they do not correlate as well with the gold standard (FOI estimates derived from seroprevalence data) as shown in Figure 1—figure supplement 2.

We have included sections in the manuscript (Materials and methods, subsection “Alternative metrics” and Results subsection, “Alternative metrics”) where we describe these results evaluating the performance of alternative metrics.

2) The authors correctly assume that all 4 serotypes could be co-endemic, and that DHF cases are likely to represent the second infection, with mild cases most likely a person's 1st-4th infection. The focus on the issues with the current vaccination recommendations is very relevant, but it is not very clear in the text whether the authors considered 'likely seropositive' after expected 1st infection (best candidate for vaccine) or whether it was any infection, with 2nd-4th not necessarily needing the vaccine as are already protected due to the homology between the strains. In epidemic-prone settings, this is somewhat simplified, but in areas with endemic transmission and co-circulating strains this is quite complex. If the model does in fact represent the likely first infection, then these nuances should be better described. If it regards the later infections, more justification for the relevance of the estimates should be included.

We agree with the reviewer that individuals who have experienced only the first dengue infection are probably the best candidates for vaccination. However, both of the SAGE recommendations (from 2016 and 2018) are based on overall seropositivity (to one or more serotypes) and do not consider the number of infections an individual may have experienced. Since one of our objectives was to generate estimates that could help countries implement the vaccine based on the WHO recommendation, we chose to focus on this metric and we used our estimates of the force of infection to calculate the expected proportion of seropositive individuals (to one or more serotypes). It would also be possible to reconstruct the expected proportion of individuals that have been infected by a single serotype, but we believe it is outside of the scope of this paper (and can’t be measured with standard serological assays).

3) To validate the estimates, the authors have plotted FOI against age-stratified serological data. However, how was case data (as labelled on the axis) estimated from FOI? Also, how were the seroprevalence estimates generated from the serological data and does this represent seropositive to all/at least 1 etc. serotypes?

We apologize for the confusion. The plot in Figure 1 shows estimates of FOI derived from case data vs estimates of FOI derived from age-stratified seroprevalence data (the gold standard). These estimates of the FOI were generated by fitting catalytic models to age-specific seroprevalence data as previously described by us and others (Ferguson et al., 1999; Rodriguez-Barraquer et al., 2011), and they represent being seroposive to at least 1 serotype.

We have updated the figure label (Figure 1C) and edited the Materials and methods subsection “Validation and sensitivity analyses” as follows:

“We validated our estimates of the force of infection by comparing them to estimates obtained from age-stratified serological data (the gold-standard) for 16 locations for which both serologic and age-specific case data was available. Methods to estimate forces of infection from seroprevalence data have been previously described”.

4) In addition to the limitations on the described method for calculating FOI, the paper would benefit from a few sentences in the Discussion on the limitations of using (very) long-term average FOI at a highly aggregate level to dictate control policy. Such measures will be insensitive to shorter-term changes (e.g., to interventions or a wide variety of factors that act at < 20-year intervals) and finer-scale heterogeneities clearly occur and clearly other metrics of transmission still have a role to play in deciding dengue control policy. Suggesting that these maps might be useful for prioritising the areas where more detailed serosurveys (or individual test and vaccinate) should be done in first instance would be a more conservative interpretation than stating that vaccinating in this particular province of Brazil is "highly questionable".

We chose to estimate average forces of infection over a 20 year period because we were interested in capturing average transmission risk for each location, rather than estimating forces of infection for specific years. We believe that these longer-term averages better reflect the transmission potential within a location and are less susceptible to transient events. However, we agree with the reviewers that averages over 20 years and over large spatial units likely conceal heterogeneities in FOI that exist at finer spatio-temporal scales.

Rather than replacing seroprevalence surveys, which are still the gold-standard, we hope that our approach and results will serve as an additional tool to help guide interventions when seroprevalence surveys are not available.

We have expanded the Discussion to incorporate both of these ideas.

“While age-stratified serological data remains the gold-standard to quantify dengue transmission, our results illustrate how inferences derived from age-specific surveillance data could be used to inform control interventions such as vaccination. […] While here we focus on presenting average forces of infection over 20 years, the same methods can also be used to estimate yearly forces of infection (Cummings et al., 2009; Hoang Quoc et al., 2016).”

“Our model also makes several assumptions that may be questionable. […] Finally, it estimates transmission intensities over extended periods of time (20 years), averaging over variations in the FOI that may occur at shorter time scales.”

5) The assumptions and limitations of the described method are well detailed in the Discussion, but it would be interesting if there were any patterns in the residuals. Are there any circumstances where this method is better or worse?

The main assumption of our model is that the age distribution of reported cases represents the age distribution of secondary infections. Thus, the main threats to model performance are biases in the reporting process that invalidate this assumption.

While it is well established that the main risk factor for dengue hemorrhagic fever is having a secondary dengue infection, it is possible that other infections (primary, tertiary or quaternary) might contribute to milder forms of the disease. Thus, the model can be expected to perform better when fit to age-specific DHF data. Using data from milder forms of dengue could further threaten model performance because “dengue fever” is a non-specific clinical diagnosis and therefore more likely to be misclassified.

As described in response to comment 6, to assess the potential impact of reporting biases we have compared FOI estimates derived from severe/DHF cases to those derived from all cases combined. These results have now been included in the manuscript.

We do not think that the residuals are a good metric of model performance because near-perfect fits do not imply unbiased estimates of the FOI. Rather than looking at the residuals, proper assessment of model performance would require comparing estimates to validation data (age-stratified serological data) from more locations.

6) A major advantage of this method is international comparability, yet (as acknowledged by the authors in the Materials and methods section, but not the Results and Discussion), it is affected by different case definitions of dengue and severe dengue used in each country. Supplementary Figure 4 shows some interesting discrepancies that are not mentioned in the Results section. Thus, how generalisable are the results in countries that have much less well-defined case definitions than the countries chosen here?

We thank the reviewer for their comment. Differences in reporting practices could certainly impact the comparability of estimates between countries. A key assumption of our model is that the age distribution of reported cases represents the age distribution of secondary infections. While it is well established that the main risk factor for dengue hemorrhagic fever is having a secondary infection, it is possible that other infections (primary, tertiary or quaternary) contribute to milder forms of the disease. Furthermore, the case definition of severe dengue and DHF is more specific, while milder forms of dengue are more likely to be misclassified.

Where possible (Thailand and Brazil) we used age-specific case reports on dengue hemorrhagic fever for our analyses. However, for both Colombia and Mexico we had to use data on all reported dengue cases because the DHF data was too sparse and a substantial number of spatial units (9/32 in Colombia and 10/31 in Mexico) didn’t include enough cases to estimate the FOI.

To assess the sensitivity of our estimates to the report type, we have compared FOI estimates derived from severe/DHF cases to those derived from all cases combined. This comparison was limited to Colombia, Brazil and Mexico because from Thailand we only had DHF data (Figure 4A). For most settings, the correlation between the estimates was good. This was particularly true for Colombia and Mexico, but less so for Brazil. However, two-thirds of the Brazilian locations that showed discrepant results had the smallest number of DHF cases (n=217 and 265) suggesting that sample size may have played a role. Similarly, all of the Colombian locations where there was a large discrepancy had very low counts of DHF cases (n= 41, 55, and 48 respectively). The state of Maranhao in Brazil showed a large discrepancy despite having large numbers of cases, due to major differences in the age distribution of severe vs. mild infections. In general, estimates derived from DHF cases tended to be slightly greater than those from all cases. However, both types of estimates correlated well with the gold standard (Figure 4B).

We think that these results are encouraging, and suggests that even data from all dengue cases can be useful to infer transmission patterns. However, it is possible that for countries that have less well-defined case definitions, it might be better to use models that make different assumptions about the observation process. For example, it may be necessary to relax the assumption that cases represent secondary infections and allow for mixtures between primary, secondary and post-secondary infections.

We have added a section on the sensitivity analyses comparing different report types (Results, subsection “Sensitivity analyses”) and added the following paragraph to the Discussion:

“There are several limitations of using age-specific surveillance data to estimate transmission parameters of dengue. […] Nevertheless, our results and sensitivity analyses are encouraging and suggest that even data from all dengue cases can be useful to infer transmission patterns.”

7) Regarding the methods: the use of additive force of infection (i.e. 3*λ) seems strange and the reviewers wonder whether it quite adds up due to the multiple permutations of ways in which a person can have a secondary infection. In a multi serotype model with equal FOI between serotypes, the probability of infection with any serotype is 1-(1-λ)^4, i.e., 1 minus probability of no infection to any serotype and slightly lower than 4*λ. This would be worth checking as it definitely affects secondary infection attack rate.

We are a bit confused by the reviewer’s comment. In our model formulation the λ is a rate (not a probability). Thus, 4*λ represents the annual rate at which susceptibles get infected by any of the four serotypes, and the annual probability of infection is 1- exp(-4*λ). Similarly, 3*λ represents the rate at which susceptibles get infected by any of 3 serotypes.

We express the expected rate of secondary infections as I(a,t)=3λ(t)z1(a,t). This represents the product of the rate of infection by 3 serotypes (because the person is already immune to the fourth one) and the probability that the person is susceptible to secondary infections. A similar approach has been previously used (Salje et al., 2019; Imai et al., 2016). The reviewer is right in pointing out that when calculating the number of secondary infections over a full year, 3*λ may slightly overestimate 1- exp(-3*λ). However, as shown in Author response image 1, this alternative formulation has no impact on our estimates of the of the force of infection.

**Author response image 1. respfig1:** Comparison of FOI estimates obtained when using the two alternative model formulations.